# On the Infinite Width and Depth Limits of Predictive Coding Networks

**Francesco Innocenti** [1 2]  **El Mehdi Achour** [3]  **Rafal Bogacz** [1 2]

## Abstract

Predictive coding (PC) is a biologically plausible alternative to standard backpropagation (BP) that minimises an energy function with respect to network activities before updating weights. Recent work has improved the training stability of deep PC networks (PCNs) by leveraging some BP-inspired reparameterisations. However, the full scalability and theoretical basis of these methods remain unclear. To address this gap, we study the infinite width and depth limits of PCNs. For linear residual networks, we show that the set of width- and depth-stable feature-learning parameterisations for PC is exactly the same as for BP. Moreover, under any of these parameterisations, the PC energy with equilibrated activities converges to the quadratic BP loss when the model width is much larger than the depth, resulting in PC computing the same gradients as BP. Experiments show that, as long as an activity equilibrium is reached, convergence to BP holds for nonlinear models including convolutional networks and transformers. Overall, this work constrains the types of parameterisation that are scalable with PC, while showing a way in which BP can be effectively implemented with only local updates in much wider than deep networks like the brain.

## 1. Introduction

How does the brain update synaptic weights using only local information available to each neuron? While backpropagation (BP) is the standard algorithm for training artificial neural networks (Rumelhart et al., 1986; LeCun et al., 2015), it is unlikely to be implemented in the brain due to its non-local propagation of information (Lillicrap et al., 2020). Understanding how the brain solves this credit assignment problem would not only greatly advance neuroscience, but could also unlock more energy efficient AI.

A more biologically plausible alternative to BP is "predictive coding" (PC), an influential theory of brain function suggesting that neurons minimise their prediction errors (Rao & Ballard, 1999; Friston, 2005). In a predictive coding network (PCN), this process occurs in two phases: first, neurons adjust their activity to minimise a sum of local objectives (or energies); then, once the network reaches an equilibrium, weights are updated to minimise the same energy function (Song et al., 2024). An advantage of PC and other local algorithms is that, at equilibrium, the weights of different layers can be updated in parallel.

While the conditions under which PC can approximate or exactly equal BP are now well-established (Whittington & Bogacz, 2017; Song et al., 2020; Millidge et al., 2022b; Rosenbaum, 2022; Salvatori et al., 2021; Millidge et al., 2022a), later work explored how standard PC could provide benefits over BP, including learning in fewer weight updates (Alonso et al., 2022; Innocenti et al., 2023; Alonso et al., 2023; Innocenti et al., 2024a) and increased performance in online and continual learning tasks (Song et al., 2024).

Encouraged by these potential advantages, recent efforts have focused on scaling PC to larger and especially deeper models (Pinchetti et al., 2024). This work has revealed, and at least partially addressed, various instabilities related to the training of deep PCNs (Qi et al., 2025; Innocenti et al., 2025; Goemaere et al., 2025). Notably, Innocenti et al. (2025) proposed a BP-derived reparameterisation of PCNs allowing stable training of 100+ layer residual networks on simple classification tasks. However, this and other reparameterisations tend to be heuristically derived, or only partially justified, and the scalability of PC to larger datasets (e.g. ImageNet) still remains to be seen.

The scaling of BP-trained models, on the other hand, is a history of success. This success is, in no small part, due to principled network parameterisations derived in "idealised" infinite width and depth limits (Yang & Hu, 2021; Bordelon & Pehlevan, 2022b; Bordelon et al., 2023; Dey et al., 2025). Such parameterisations enable not only stable training dynamics across model sizes, but also "zero-shot hyperparameter transfer" (Yang & Hu, 2021), whereby optimal tuning parameters such as the learning rate remain

[1]Brain Network Dynamics Unit, University of Oxford, UK [2]MRC CoRE in Restorative Neural Dynamics, UK [3]UM6P College of Computing, Rabat, Morocco. Correspondence to: Francesco Innocenti <francesco.innocenti@ndcn.ox.ac.uk>.

*Proceedings of the 43rd International Conference on Machine Learning*, Seoul, South Korea. PMLR 306, 2026. Copyright 2026 by the author(s).

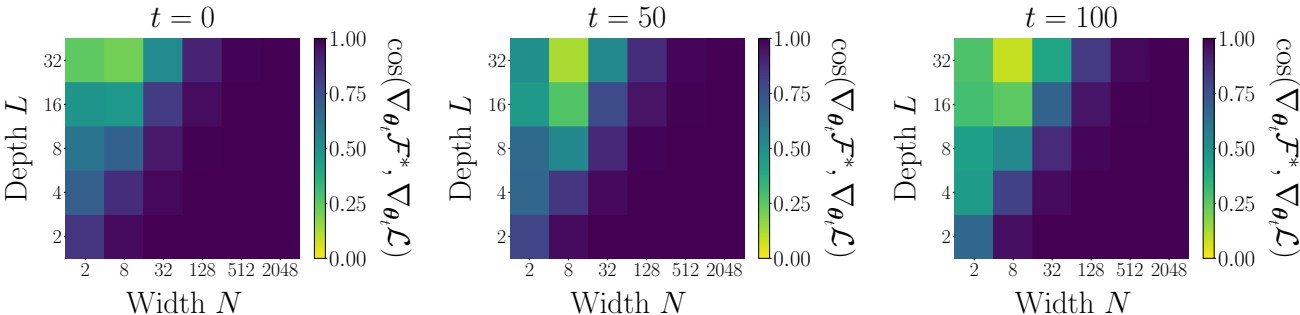

*Figure 1.* **Under width- and depth-stable feature-learning parameterisations of linear residual networks, PC converges to BP when the model width is much larger than the depth,** $N \gg L$. We trained linear residual networks on CIFAR-10 with the mean-field parameterisation (as defined in Table 2) and depth scaling exponent $\alpha = 1/2$ (§4). Plotted are the cosine similarities averaged over 3 random runs between the equilibrated energy (Eq. 5) gradients (PC) and the MSE loss (Eq. 1) gradients (BP) at different training steps $t$. See §A.8 for more details and Figure A.6 for similar results on Fashion-MNIST.

approximately constant across scales.

Moreover, analyses of infinitely wide networks have already provided insights into the learning dynamics of other bio-plausible learning rules (Bordelon & Pehlevan, 2022a). Given this, and the practical impact of theory-driven parameterisations for BP (Yang & Hu, 2021)—which have already been shown to improve the scalability of PCNs (Innocenti et al., 2025)—this leads us to the following question:

*Can we derive principled parameterisations specific to PC that allow stable and more efficient training of both wide and deep networks?*

Here, we positively answer this question by analysing the infinite width and depth limits of linear PCNs, with supporting experiments on nonlinear models. Surprisingly, we prove that the set of width- and depth-stable feature-learning ("non-lazy") parameterisations for PC is exactly the same as for BP. Moreover, under any of these parameterisations, we show that the weight gradients computed by PC converge to the BP gradients in a regime where the model width is much larger than the depth.

We demonstrate that these theoretical results hold in practice for nonlinear models, as long as an equilibrium of the activities is reached. We perform experiments with different architectures including convolutional neural networks (CNNs) and transformers, trained with different nonlinearities, optimisers and loss functions. Overall, our work provides hard constraints on the types of parameterisation that are scalable with PC, while showing a way in which BP can be effectively implemented with a local algorithm (and perhaps by the brain) in much wider than deep networks.

The rest of the paper is organised as follows. After some background on PCNs and parameterisations for BP, Section 3 presents our results on the width parameterisations of linear multi-layer perceptrons (MLPs) for PC. This is fol-

lowed by results on the depth PC parameterisation of linear residual networks (§4). We then present experiments with nonlinear models supporting our theoretical results (§5), before concluding with the implications and limitations of this work (§6). For space reasons, we refer related work, additional experiments and derivations to Appendix A.

### 1.1. Summary of contributions

- For linear MLPs with equilibrated activities, we show that the set of width-stable and feature-learning parameterisations for PC is the same as for BP (Theorem 1). In addition, under any of these parameterisations, PC computes the same gradients as BP for sufficiently large width (Corollary 3.2; Figures 2 & A.15).

- We generalise these results to linear residual networks, proving that the set of width- *and* depth-stable feature-learning parameterisations is also equivalent for PC and BP (Theorem 2; see also Figure A.5). Similar to the MLP case, the PC gradients converge to BP's in a regime where the model width is much larger than the depth (Corollary 4.2; Figures 1 & A.6).

- We empirically verify our theoretical results on linear and nonlinear architectures including CNNs and transformers (e.g. Figure 4), trained on toy and large-scale tasks (e.g. ImageNet), with different optimisers and loss functions. We also provide heuristic extensions of our derived PC parameterisations for CNNs, transformers, and the Adam optimiser (§A.6).

## 2. Background

To set up our theoretical analysis of PCNs (§3), in this section we review PC, define our general parameterisation, and revisit "width-aware" parameterisations for BP. For general notation, see §A.1.

## 2.1. Predictive coding networks

We consider the standard supervised learning setting given a dataset of $P$ examples $\mathcal{D} = \{(\mathbf{x}_\mu, y_\mu)\}_{\mu=1}^P$, with $\mathbf{x}_\mu \in \mathbb{R}^D$ and scalar targets $y_\mu \in \mathbb{R}$, which we assume for simplicity throughout. For comparison, it is useful to recall the standard mean squared error (MSE) loss

$$\mathcal{L}(\boldsymbol{\theta}) = \frac{1}{2P} \sum_{\mu=1}^P (y_\mu - f(\mathbf{x}_\mu; \boldsymbol{\theta}))^2, \qquad (1)$$

where $f(\mathbf{x}_\mu; \boldsymbol{\theta})$ denotes a model's prediction for a given input, with parameters $\boldsymbol{\theta} \in \mathbb{R}^p$.

**PC energy.** PCNs minimise a sum of local objectives (energies) that typically take the form of layer-wise MSEs (Buckley et al., 2017; Bogacz, 2017). For concreteness, consider the PC energy of a simple MLP with no biases

$$\mathcal{F}(\mathbf{z}, \boldsymbol{\theta}) = \frac{1}{2P} \sum_{\mu=1}^P \sum_{\ell=1}^L \left\| \mathbf{z}_\mu^{(\ell)} - \phi_\ell(\mathbf{W}^{(\ell)} \mathbf{z}_\mu^{(\ell-1)}) \right\|^2, \quad (2)$$

where $\mathbf{z}_\mu := \{\mathbf{z}_\mu^{(\ell)}\}_{\ell=0}^L$ denote extra (latent) variables that are potentially free to vary, $\{\mathbf{W}^{(\ell)}\}_{\ell=1}^L$ are weight matrices, and $\phi_\ell(\cdot)$ is a layer activation function.

**PC inference & learning.** For supervised learning, PCNs are trained by clamping the first and last layer to some input and target data: $z_\mu^{(L)} \leftarrow y_\mu$ and $\mathbf{z}_\mu^{(0)} \leftarrow \mathbf{x}_\mu$. The energy (Eq. 2) is then minimised in a bi-level, expectation-maximisation fashion. First, given some weights $\boldsymbol{\theta}_t$, we minimise the energy with respect to the activities of the network,

$$\mathbf{z}_\mu^* = \arg\min_{\mathbf{z}_\mu} \mathcal{F}(\mathbf{z}_\mu, \boldsymbol{\theta}_t). \qquad (3)$$

This process is called "inference" and can be intuitively thought as the network trying to find an equilibrium of its state that best accounts for each data sample. Eq. 3 is typically minimised using standard gradient descent (GD): $\mathbf{z}_{k+1} = \mathbf{z}_k - \beta \nabla_{\mathbf{z}} \mathcal{F}$ with some step size $\beta$, where for simplicity we drop the data index $\mu$. After convergence of the activities $\mathbf{z}_\mu^*$, we minimise the energy with respect to the weights, by performing a single (e.g. GD) update:

$$\boldsymbol{\theta}_{t+1} = \boldsymbol{\theta}_t - \eta \frac{\partial \mathcal{F}(\mathbf{z}^*, \boldsymbol{\theta}_t)}{\partial \boldsymbol{\theta}}, \qquad (4)$$

where $\eta$ is some learning rate. The optimisation cycle is

then restarted with a new data batch. Depending on the task (e.g. discrimination vs generation), the network can be tested to predict a target given some input, or to infer an input given some target. In contrast to BP, both the activity and weight gradients of the energy for any layer are local, in that they only require information from adjacent layers.

**Equilibrated energy for linear networks.** For arbitrary linear networks with $\phi_\ell(\cdot) = \mathbf{I}$ for all $\ell$, the inference minimisation problem of Eq. 3 has a unique solution $\mathbf{z}_\mu^*(\boldsymbol{\theta})$ where $\partial\mathcal{F}/\partial\mathbf{z}_\mu^* = \mathbf{0}$ (Ishikawa et al., 2024; Innocenti et al., 2025). Moreover, as shown by Innocenti et al. (2024a), the PC energy (Eq. 2) evaluated at this solution $\mathcal{F}(\mathbf{z}_\mu^*(\boldsymbol{\theta}), \boldsymbol{\theta})$—which we will refer to as the *equilibrated energy* and abbreviate as $\mathcal{F}^*(\boldsymbol{\theta})$—is equivalent to a rescaled MSE loss with a non-trivial, weight-dependent rescaling. For a linear MLP with scalar output, this takes the following form:

$$\mathcal{F}^*(\boldsymbol{\theta}) = \frac{1}{s(\boldsymbol{\theta})} \mathcal{L}(\boldsymbol{\theta}), \qquad (5)$$

$$s(\boldsymbol{\theta}) = 1 + \sum_{\ell=2}^L \|\mathbf{W}^{(L:\ell)}\|^2, \qquad (6)$$

where we define the product $\mathbf{W}^{(L:\ell)} := \mathbf{w}^{(L)}\mathbf{W}^{(L-1)}\dots\mathbf{W}^{(\ell)}$ for $\ell \in \{1,\dots,L\}$. For intuition, the rescaling for one hidden layer simplifies to a squared norm of the output weights plus one, $s(\boldsymbol{\theta}) = 1 + \|\mathbf{w}^{(L)}\|^2$. Importantly, Eq. 5 is the objective on which PC effectively learns for linear networks, and insights obtained from the equilibrated energy have been empirically shown to transfer to nonlinear PCNs (Innocenti et al., 2024a). For this reason, Eq. 5 will be the basis of all our theoretical results.

## 2.2. General parameterisation

We begin by studying linear MLPs of width $N$ and depth $L$ with scalar output. We will empirically show that our theory holds for multidimensional output and different nonlinear networks. Our setting and notation are closest to those of Bordelon & Pehlevan (2022b; 2025). In particular, we consider the following general MLP parameterisation:

$$f_\mu = \frac{1}{\gamma} h_\mu^{(L)}, \qquad (7)$$

$$h_\mu^{(L)} = \frac{1}{N^{a_L}} \mathbf{w}^{(L)} \mathbf{h}_\mu^{(L-1)}, \qquad (8)$$

$$\mathbf{h}_\mu^{(\ell)} = \frac{1}{N^{a_\ell}} \mathbf{W}^{(\ell)} \mathbf{h}_\mu^{(\ell-1)}, \qquad (9)$$

$$\mathbf{h}_\mu^{(1)} = \frac{1}{N^{a_1}\sqrt{D}} \mathbf{W}^{(1)} \mathbf{x}_\mu, \qquad (10)$$

*Table 1.* Width-dependent scaling exponents for MLPs of width $N$

| Exponent | Scaling term | Description |
|---|---|---|
| $a_\ell$ | Preactivation | Determines the $N^{-a_\ell}$ scaling of the layer preactivations. |
| $b_\ell$ | Init. variance | Determines the $N^{-b_\ell}$ scaling of the weight init. variance. |
| $c$ | Learning rate | Part of $\eta = \eta_0 \gamma^2 N^{-c}$, scaling the global learning rate. |
| $d$ | Network output | Part of $\gamma = \gamma_0 N^d$, scaling the network predictions. |

with layer activations[1] $\mathbf{h}_\mu^{(\ell)} \in \mathbb{R}^N$ for $\ell \in \{1, \ldots, L-1\}$, and weight matrices $\mathbf{W}^{(1)} \in \mathbb{R}^{N \times D}$, $\mathbf{W}^{(\ell)} \in \mathbb{R}^{N \times N}$ and $\mathbf{w}^{(L)} \in \mathbb{R}^{1 \times N}$. Biases are ignored for simplicity. The exponents $a_\ell \in \mathbb{R}$ determine how the activations are scaled with the network width $N$. As we will see, the output scaling

$$\gamma = \gamma_0 N^d \qquad (11)$$

determines the learning regime (rich vs lazy), with $\gamma_0$ as a width-independent constant controlling the strength or "richness" of feature learning, and $d$ as a width scaling exponent. All the weights are initialised i.i.d. as

$$W_{ij}^{(\ell)} \sim \mathcal{N}(0, N^{-b_\ell}), \quad \ell \in \{1, \ldots, L\}, \qquad (12)$$

where $b_\ell$ determines how the variance is scaled with the width. Note that the "standard parameterisation" (SP) used in practice (e.g. in PyTorch) assumes $b_\ell = 1$ for $\ell > 1$ and $a_\ell = 0$ for all $\ell$, since standard initialisations (LeCun et al., 2002; He et al., 2015) ensure that weights have variance inversely proportional to their input dimension. Lastly, we will in the first instance consider gradient flow (i.e. continuous-time GD)

$$\frac{d\boldsymbol{\theta}}{dt} = -\eta \frac{\partial l(\boldsymbol{\theta})}{\partial \boldsymbol{\theta}}, \qquad (13)$$

where depending on the context, $l(\boldsymbol{\theta})$ will denote the MSE loss (Eq. 1) or the equilibrated energy (Eq. 5). Our results are straightforward to generalise to other optimisers, and we provide a simple extension for Adam (Kingma & Ba, 2014) in §A.6. Following Bordelon & Pehlevan (2022b), the learning rate is parameterised as

$$\eta = \eta_0 \gamma^2 N^{-c}, \qquad (14)$$

where $\eta_0$ is a constant, the $\gamma^2$ factor is included to make the change in the loss at initialisation independent of $\gamma$ (see §A.4.3), and $c$ is another width scaling exponent.

A (width-aware) parameterisation is simply defined by the set of width-dependent scaling exponents $(a_\ell, b_\ell, c, d)$, as summarised in Table 1. We will extend this to depth in §4. Following the analysis done for BP (reviewed in the next section), our goal will be to derive parameterisations (i.e. solve for the exponents) for PC under specific constraints.

## 2.3. Width-aware parameterisations for BP

Having defined our general "abcd" parameterisation (§2.2), we now review the influential notion of a "**stable and feature-learning parameterisation**", also known as the "maximal update" ($\mu$P) (Yang & Hu, 2021) or "mean-field" (Bordelon & Pehlevan, 2022b) parameterisation. This type of parameterisation is defined by satisfying 3 main desiderata that are aimed at ensuring both stable and non-trivial learning dynamics in the infinite-width limit (Yang & Hu, 2020; 2021; Bordelon & Pehlevan, 2022b).

> **Parameterisation desiderata at large width $N$.**
>
> - **Desideratum 1.** Layer preactivations are stable with respect to the width at initialisation: $h_i^{(\ell)} = \Theta_N(1)$.
>
> - **Desideratum 2.** Network predictions are width-stable during training: $df/dt = \Theta_N(1)$.
>
> - **Desideratum 3.** Layer features or preactivations are also width-stable during training: $dh_i^{(\ell)}/dt = \Theta_N(1)$.

Here $h_i^{(\ell)}$ refers to the $i$th neuron in the $\ell$th layer, and the notation $x = \Theta_N(1)$ indicates that $x$ neither vanishes nor explodes (it is hence "stable") with $N$ (for a precise definition, see §A.1).[2] The first two constraints relate to the numerical stability of the model with the width, while the last desideratum characterises the degree to which the features evolve during training. Parameterisations that violate (1) or (2) are therefore commonly said to be "unstable", while those that do not learn features are labelled as trivial or "lazy". For BP, there is a known set of one-dimensional parameterisations that satisfy all 3 desiderata (Yang & Hu, 2021; Bordelon & Pehlevan, 2022b):

$$\begin{cases} 2a_\ell + b_\ell = 1 & \text{for} \quad \ell \in \{2, \ldots, L\}, \quad 2a_1 + b_1 = 0, \\ 2a_\ell + c = 1 & \text{for} \quad \ell \in \{2, \ldots, L\}, \quad 2a_1 + c = 0, \\ d = 1/2. \end{cases}$$

$$(15)$$

That is, if one fixes any among $(a_\ell, b_\ell, c)$, then there is a

---

[1]Note that for linear networks there is no distinction between pre- and post-activations.

[2]This is equivalent to requiring that the average squared (Euclidean) norm of each layer preactivation is $\frac{1}{N}\|\mathbf{h}^{(\ell)}\|^2 = \Theta_N(1)$.

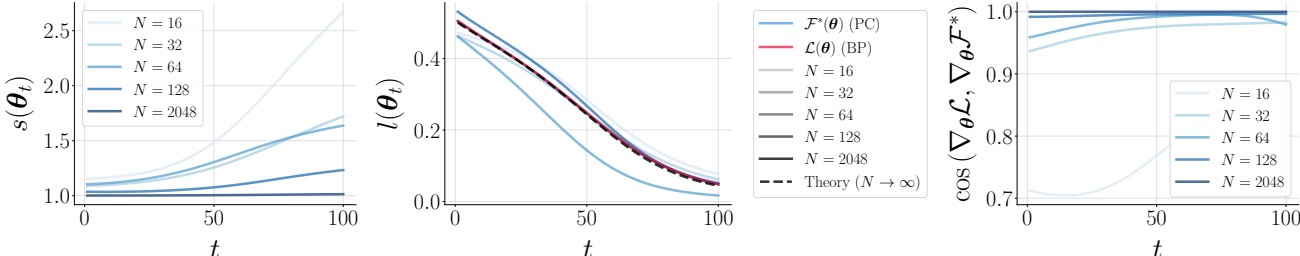

*Figure 2.* **Under width-stable and feature-learning parameterisations of linear MLPs, PC converges to BP at large width.** We trained deep linear MLPs ($L = 5$) of varying widths $N$ with full-batch GD on a toy task with binary labels. All models used the mean-field parameterisation as defined in Table 2. For comparative results with the SP, see Figure A.19. (*Left*) As predicted by Eq. 16, the equilibrated energy rescaling $s(\theta)$ approaches one as $N \to \infty$. (*Middle*) As a result, the equilibrated energy $\mathcal{F}^*(\theta)$ (Eq. 5) converges to the MSE loss $\mathcal{L}(\theta)$ (Eq. 1), and PC effectively computes the same gradients as BP (*Right*). The theoretical loss was calculated using dynamical mean field theory (see §A.8 for more details). For additional results including on CIFAR-10, see Figures A.18 & A.15.

unique solution. For example, if one chooses $a_\ell = 1/2$ for $\ell > 1$, then one obtains the mean-field parameterisation considered by Bordelon & Pehlevan (2022b), with $b_\ell = c = a_1 = 0$. $\mu$P as introduced in Yang & Hu (2021) further requires that the learning rate $\eta$ is $\Theta_N(1)$, leading to slightly different scalings. Note that the feature learning desideratum (3) is uniquely satisfied by $d = 1/2$, and this is the sense in which $\gamma$ (Eq. 11) is said to determine the learning regime.

For comparison, note that the SP has stable initialisation as well as evolving features, but the network predictions explode with the width, as shown by Yang & Hu (2020). On the other hand, the so-called neural tangent kernel (NTK) parameterisation (Jacot et al., 2018) is fully stable (in that it satisfies the first 2 constraints), but its feature updates vanish with the width (hence known as "lazy"). For the specific scalings defining all these parameterisations, see Table 2.

## 3. Width-aware Parameterisations for PC

Given the empirical success of the width-stable and feature-learning BP parameterisations, we might want to ensure the same desiderata (listed in §2.3) for PC. It turns out that, at least for linear networks with equilibrated activities, PC admits exactly the same set of parameterisations as BP.

**Theorem 1** (Width-stable and feature-learning parameterisations for linear PCNs). *Consider the $(a_\ell, b_\ell, c, d)$ parameterisation of linear MLPs (Eqs. 7-10), and assume PCNs with converged activities that therefore learn on the equilibrated energy (Eq. 5). Then, there exists a set of one-dimensional parameterisations that satisfies the 3 Desiderata in §2.3 for PC, and this is the same as for BP (Eq. 15).*

See Figure A.5 for a visual illustration of the result. The full derivation is provided in §A.4 and involves checking every desideratum for linear PCNs performing gradient flow

on the equilibrated energy (Eq. 5). Verification of the first desideratum does not differ between BP and PC (as the forward pass is algorithm-independent), while the other constraints involve taking into account extra terms arising from the rescaling of the equilibrated energy (Eq. 6).

As an intuition for Theorem 1, one can show that, as for BP, there is a unique output scaling exponent $d = 1/2$ that ensures rich (non-lazy) feature learning during training of PCNs. As we will see next, this scaling causes the equilibrated energy terms to become subleading compared to the loss terms, vanishing at a rate of $\Theta(N^{-1/2})$.

### 3.1. PC convergence to BP for linear MLPs as $N \to \infty$

We just showed that the set of width-stable and feature-learning parameterisations for PC is the same as for BP (Theorem 1). Under any of these parameterisations, it is straightforward to show that the weight gradients computed by PC will converge to BP's in the infinite-width limit.

**Corollary 3.2** (PC convergence to BP on wide linear MLPs.). *Under any width-stable and feature-learning parameterisation of linear MLPs (Eq. 15), the equilibrated energy (Eq. 5) converges to the MSE loss (Eq. 1) as $N \to \infty$, resulting in PC computing the same gradients as BP.*

For intuition, consider the equilibrated energy rescaling (Eq. 6) for one-hidden-layer PCNs under the mean-field parameterisation (see Table 2): $s(\theta) = 1 + \gamma_0^{-2} N^{-2} \|\mathbf{w}^{(L)}\|^2$. Given the first desideratum's constraint (§2.3), the norm term scales as $\|\mathbf{w}^{(L)}\|^2 = \Theta(N)$, leading to

$$s(\theta) = 1 + \frac{1}{\gamma_0^2 N} \Theta_N(1) = 1 + \Theta(N^{-1}) \quad (16)$$

$$\implies \quad \mathcal{F}^*(\theta) \to \mathcal{L}(\theta), \quad \text{as } N \to \infty. \quad (17)$$

We see that the energy rescaling approaches one in the

infinite-width limit, which implies that the equilibrated energy (Eq. 5) converges to the MSE loss (Eq. 1), and PC computes the same gradients as BP. All these results are empirically verified in Figures 2, A.15 & A.21 for both linear and nonlinear networks trained on simple tasks. The result of Eq. 16 can be generalised to more layers by noting (i) that each squared norm term in Eq. 6 is $\Theta_N(1)$ under the forward pass desideratum (see Eq. 46), and (ii) that $\gamma^2$ scales every term in the sum of Eq. 6. We note that a closely related result was proved by Ishikawa et al. (2024), which we discuss in more detail in §6 and §A.2.

As a further intuition for why the parameterization of Eq. 15 results in the PC weight gradients converging to BP's, note that the effective output weights are scaled by an additional $1/\sqrt{N}$ factor compared to the hidden weights (as $d = 1/2$). Under this condition, the activity of the hidden layers after inference (Eq. 3) becomes close to that determined by the forward pass, i.e. $\mathbf{z}^{*(\ell)} \approx \mathbf{h}^{(\ell)}$. Hence, the error terms for $\ell < L$ vanish, the PC energy (Eq. 2) approaches the loss (Eq. 1), and the PC gradients approximate BP's, as first shown by Whittington & Bogacz (2017).

> **Takeaway 1**: *The set of width-stable and feature-learning parameterisations for linear PCNs with equilibrated activities is exactly the same as for BP. Moreover, under any of these parameterisations, PC converges to BP for sufficiently large width.*

### 3.2. Learning regimes

**BP regimes.** Since PC converges to BP under any width-stable feature-learning parameterisation as $N \to \infty$ (Corollary 3.2), it inherits BP's known learning regimes. The output constant $\gamma_0$ (Eq. 11) can be used to interpolate between these different regimes:

- $\gamma_0 < 1$: PC will exhibit kernel (lazy) behaviour.

- $\gamma_0 = 1$ is equivalent to $\mu$P and recovers the parameterisation introduced by Ishikawa et al. (2024) for PCNs up to one $(a_\ell, b_\ell, c)$ degree of freedom.

- $\gamma_0 > 1$ leads to richer feature learning, with $\gamma_0 \to \infty$ recovering the small initialisation or saddle-to-saddle regime (Saxe et al., 2013; Jacot et al., 2021).

Figure A.3 verifies these regimes for PC, showing that larger values of $\gamma_0$ are associated with faster learning. We emphasise that these learning regimes were established for BP and so are not specific to PC. However, next we show how our general parameterisation can provide insights into a previously identified regime that is unique to PC.

**Fast (but unstable) saddle-to-saddle PC regime.** Recently, Innocenti et al. (2024a) showed that many degenerate saddle points of the MSE loss (Eq. 1) become benign in the equilibrated energy (Eq. 5), and so much easier to escape. For deeper than wide models ($L > N$), this leads to significantly faster saddle-to-saddle dynamics for PC than BP for small initialisation (and SGD with small learning rate) (Innocenti et al., 2024a). Such a regime can be recovered by noting that deeper than wide networks under the SP, as assumed by Innocenti et al. (2024a), are effectively initialised near the origin (see Figure A.4).

However, this fast PC saddle regime is unstable in two distinct ways. First, the more one would like to increase $N$, the more $L$ would have to grow to preserve the PC speed-ups, as shown in Figure A.4. However, because this regime relies on the SP, the model is unstable at large width, in the sense of violating Desideratum 2 (§2.3). Moreover, this regime also grows unstable with the depth, in that the variance of the MLP preactivations is prone to vanish/explode except under highly restrictive conditions (Poole et al., 2016; Schoenholz et al., 2016) (see also Figures A.7 & A.13).

As we will see next, residual models can make the forward pass much more stable with the depth. However, as noted by Innocenti et al. (2024a), residual networks do not exhibit the same saddle regime as MLPs, since they effectively shift the position of the origin saddle (Hardt & Ma, 2016). We therefore reach the following conclusion.

> **Takeaway 2**: *The only regime where PC has been shown to provide potential benefits over BP in the form of "fast saddle-to-saddle dynamics" grows unstable with both the model width and depth.*

## 4. Depth-aware Parameterisations for PC

Following previous work for BP (Yang et al., 2023; Bordelon et al., 2023), we now extend our width PC parameterisations for MLPs (§3) to depth for residual networks, which have a preactivation variance that can be more easily controlled with the depth (see §A.5.1). For simplicity, let us fix a specific width-stable feature-learning parameterisation with $a_\ell = 1/2$ for $\ell > 1$ (see Table 2). To extend our previous MLP parameterisation (§2.2), we simply redefine the layer activations for $\ell \in \{2, \dots, L-1\}$ (Eq. 9) as

$$\mathbf{h}_\mu^{(\ell)} = \left( \mathbf{I} + \frac{1}{L^\alpha \sqrt{N}} \mathbf{W}^{(\ell)} \right) \mathbf{h}_\mu^{(\ell-1)}, \qquad (18)$$

where we introduced a (one-block) skip connection and a depth-dependent scaling exponent $\alpha$. In such a parameterisation, it is typical to require the following two depth-specific desiderata (Yang et al., 2023; Bordelon et al., 2023).

**Parameterisation desiderata at large depth $L$.**

- **Desideratum 4.** Layer preactivations are depth-stable at initialisation: $h_i^{(\ell)} = \Theta_L(1)$.

- **Desideratum 5.** Layer features are also depth-stable during training: $dh_i^{(\ell)}/dt = \Theta_L(1)$.

As for the width (§2.3), $x = \Theta_L(1)$ means that $x$ is constant with respect to the depth $L$. As shown in previous work (Yang et al., 2023; Bordelon et al., 2023), the residual activations (Eq. 18) explode—and so Desideratum 4 is violated—unless $\alpha \geq 1/2$ (see §A.5.1). Note that this includes the SP, where $\alpha = 0$. Further, one can show that $\alpha = 1/2$ uniquely satisfies Desideratum 5. As for the width case (Theorem 1), this also turns out to be the only parameterisation that satisfies all the depth desiderata for PC.

> **Theorem 2** (Width- and depth-stable feature-learning parameterisations for linear PCNs.). *Consider any width-stable and feature-learning parameterisation of linear residual networks (Eq. 15) with an additional depth scaling exponent $\alpha$ (Eqs. 7-8, 18 & 10). Assume PCNs that learn on the equilibrated energy (with rescaling as in Eq. 104). Then, the parameterisation that satisfies the depth Desiderata for PC is the same as for BP (i.e. $\alpha = 1/2$).*

The result is proved in §A.5. Similar to the width case, the forward pass desideratum (4) is exactly the same for BP and PC. We note that Innocenti et al. (2025) used a version of this parameterisation to train 100+ layer residual networks with PC. However, it was only justified by the forward pass stability with depth. Our result therefore provides a more solid justification for this parameterisation (see also §A.2).

### 4.1. PC convergence to BP for linear residual networks when $N \gg L$

Similar to the MLP case (§3), one can show that under any width- *and* depth-stable feature-learning parameterisation of linear residual networks, PC converges to BP in a regime where the model width is much larger than the depth.

> **Corollary 4.2** (PC convergence to BP on deep and wide linear residual networks.). *Under any width- and depth-stable feature-learning parameterisation of linear residual networks (Eq. 15) with $\alpha = 1/2$, the equilibrated energy (Eq. 5) converges to the MSE loss (Eq. 1) when $N \gg L$, resulting in PC computing the same gradients as BP.*

This is easy to see once one realises that, in the infinite

width and depth limit, the equilibrated energy rescaling of linear residual networks (Eq. 104) scales as

$$s(\boldsymbol{\theta}) = 1 + \Theta\left(L/N\right), \quad \text{as } N, L \to \infty. \quad (19)$$

This scaling is empirically verified in Figures 3 and A.9. Thus, similar to the MLP case, as long as $N \gg L$, (i) the rescaling approaches one, (ii) the equilibrated energy converges to the MSE loss, and (iii) the PC gradients converge to the BP gradients. All these results are verified for both linear and nonlinear residual networks (e.g. see Figures 1 & A.11). We note that Corollary 4.2 provides a non-trivial generalisation of a result of Innocenti et al. (2025), which proved this only at initialisation (see §A.2).

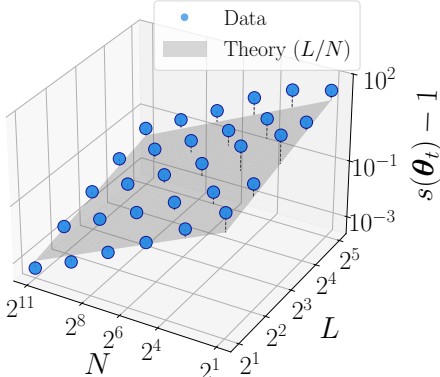

*Figure 3.* **Empirical verification of Eq. 19.** For linear residual networks trained on CIFAR-10, we plot the empirical equilibrated energy rescaling $s(\boldsymbol{\theta})$ (Eq. 104) minus one as a function of the width $N$ and depth $L$, against the $L/N$ theoretical prediction (Eq. 19). Note that the same scaling applies to MLPs with infinite depth (Figure A.10), but the stability of their forward pass with depth is more fragile, as discussed in §3.2.

> *Takeaway 3: The set of width- and depth-stable feature-learning parameterisations for PC is the same as for BP. Moreover, under any of these parameterisations, PC converges to BP as long as the width is much larger than the depth.*

## 5. Experiments with Practical PCNs

Our theoretical analysis makes two important related assumptions: (i) linear networks, and (ii) exact convergence of the PC inference dynamics (Eq. 3). Together, these assumptions allow us to study the equilibrated energy (Eq. 5) as the effective energy on which (linear) PCNs learn. To avoid convergence failure as a potential confounder in testing our theory, all previous experiments performed GD directly on the equilibrated energy. However, in practice we are interested in nonlinear networks, for which there is in general no analytical solution of the activities, and inference is performed by some iterative algorithm such as GD (§2.1).

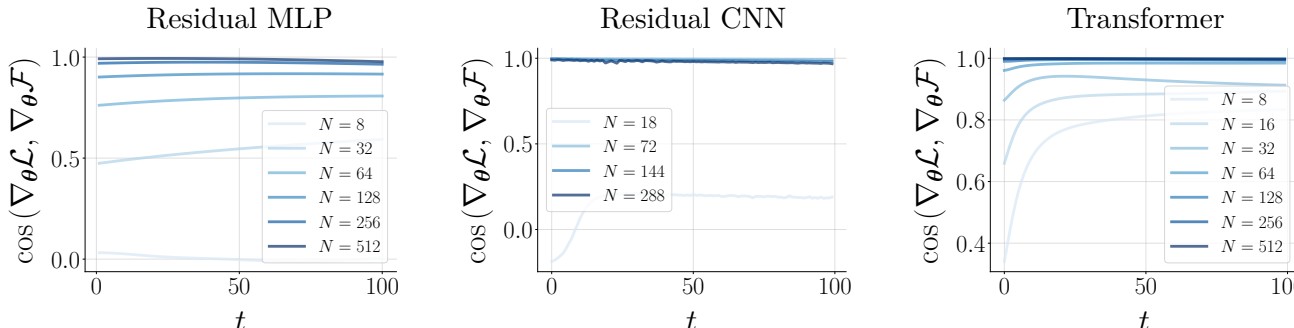

*Figure 4.* **PC also converges to BP on different *nonlinear architectures* that are much wider than deep, under stable and feature-learning parameterisations.** We trained residual MLPs, residual CNNs, and (nano-GPT-style) transformers with appropriate extensions of the $\mu$P parameterisation (§A.6). The architecture and training details for the residual MLPs were the same as the 32-layer models in Figure 1, with the addition of Tanh as nonlinearity. The residual CNNs had 10 layers, an effective width of $N = k^2 C_{\text{out}}$ with kernel size $k = 3$ and output channels $C_{\text{out}} \in \{2, 8, 16, 32\}$, and were trained on ImageNet-1k using the cross-entropy loss. The transformers had 12 blocks with 8 attention heads, and were trained on character-level Tiny Shakespeare. For more details, see §A.6 and §A.8. Plotted are the cosine similarities for a representative run between the BP loss gradients and the energy gradients at the last step of activity (GD) optimisation (Eq. 3), for different training steps $t$. Results were similar across seeds.

This raises the question: *under any stable and feature-learning parameterisation, does PC still converge to BP in practice when $N \gg L$, i.e. on nonlinear networks when performing inference via optimisation?*

To answer this question, we trained different nonlinear architectures, including CNNs and (nano-GPT-style) transformers (Radford et al., 2019). All experiments used Adam and the mean-field parameterisation, with heuristic extensions for different architectures (see §A.6 & §A.8 for details). Code to reproduce all the results is available at https://github.com/thebuckleylab/jpc/tree/main/experiments/limits_paper.

Remarkably, across all these settings, we find that PC still reliably converges to BP for wider than deep models (Figure 4), as long an an activity equilibrium seems to be reached. For deeper networks, we found that convergence held only given a large number of inference steps, in line with previous work (Innocenti et al., 2025; Goemaere et al., 2025). These results clearly show that our linear theory remains valid for nonlinear models.

> **Takeaway 4**: *Under stable and feature-learning parameterisations, PC still converges to BP on different nonlinear architectures that are much wider than deep, as long as an activity equilibrium seems to be reached.*

## 6. Discussion

### 6.1. Summary

We showed that the set of stable and feature-learning parameterisations with respect to both model width and depth for PC is exactly the same as for BP (Theorems 1-2; Fig-

ure A.5). Under any of these parameterisations, the weight gradients computed by PC converge to the BP gradients when the model is much wider than deep (and the depth is not too large for MLPs) (Corollaries 3.2 & 4.2). Finally, we showed that even in practice, when performing inference via optimisation and on nonlinear networks, PC still converges to BP for $N \gg L$ under such parameterisations, as long as an activity equilibrium is reached (Figure 4).

### 6.2. Implications

Overall, our results provide hard constraints on the types of parameterisation that are scalable (in model width and depth) with PC. More explicitly, we showed that

> *if one would like to satisfy the $\mu$P desiderata (§2.3 & §4), then **necessarily** the $\mu$P/mean-field parameterisation is the **only** scalable parameterisation for PC (Theorems 1-2; Figure A.5), in the sense of being numerically stable and learning non-trivial features at large width and depth.*

These results go significantly beyond the comparable Corollary 4.3 of Ishikawa et al. (2024), which only showed that there *exists* a parameterisation where linear PCNs implement BP. Also note that Theorems 1-2 imply that PCNs commonly trained in practice are inherently unstable with respect to both width and depth, since they rely on the SP. As discussed in §3.2, this includes the unique regime where PC has demonstrated clear benefits over BP in the form of fast saddle-to-saddle dynamics (Innocenti et al., 2024a). However, we cannot necessarily conclude from these results that there exists no other notion of a stable and rich parameterisation under which PC does not converge to BP. We return to this point below.

While the above results may appear negative, they are also the first (to the best of our knowledge) to show—theoretically for linear networks and empirically for nonlinear ones (Figure 4)—that

> *BP can be effectively implemented with a local algorithm (PC) **at scale**, for models that are much wider than deep.*

This result is encouraging for two reasons. First, it aligns with modern large language models, whose width typically exceeds their depth by at least one order of magnitude. This means that, if the convergence of PC inference (Eq. 3) could be substantially accelerated (a point which we address below), then training of BP models could be sped up by a factor of $L$, since the weight updates of PC are parallelisable across layers. Second, the brain is in fact much wider than deep (Suzuki et al., 2023), with around $5 - 10$k synapses per neuron and 6 cortical layers. Our results therefore suggest a way in which biology could implement BP with a purely local learning rule.

### 6.3. Limitations and future directions

**Linear theory.** While we provided supporting experiments on nonlinear models (Figure 4), linearity remains the main limitation of our theoretical analysis. Building further on Bordelon & Pehlevan (2022b), it could be interesting to investigate whether our theoretical results can be extended to the nonlinear case by using tools from dynamical mean-field theory (DMFT). DMFT allows one to derive a lower-dimensional description of network dynamics in certain limits, and has already been used to gain insights into infinite-width networks trained with other biologically plausible rules, including feedback alignment and error-modulated Hebbian learning (Bordelon & Pehlevan, 2022a). These algorithms do not have alternating activity and weight dynamics like PC, but one could imagine deriving DMFT equations for both optimisation timescales.[3]

**Alternative parameterisations.** Another interesting direction would be to investigate other types of parameterisation (§2.2), with at least two goals. First, as mentioned above, there could be other notions of a stable and feature-learning parameterisation (see Figure A.5 for a schematic illustration), where PC might not not converge to BP. Exploring such other notions would require considering different desiderata (§2.3 & §4). It could also be useful to consider constraints for stable inference, as well as learning, dynamics. This was in part attempted in Ishikawa et al. (2024) by considering scalings of the layer-wise PC energies. Similarly motivated, Innocenti et al. (2025) suggested that the conditioning of the inference landscape should remain con-

stant with respect to the width and depth, but found that this desideratum was fundamentally at odds with the stability of the forward pass with depth (§4).

**PC inference cost.** The main limitation of PC remains the computational cost of inference convergence (Eq. 3), shared with other bio-plausible algorithms with alternating dynamics such as equilibrium propagation (Scellier & Bengio, 2017). On GPUs, it has been estimated that PC becomes about as fast as BP when the number of inference steps is approximately equal to the depth (Salvatori et al., 2024), since as previously noted weight updates are layer-parallelisable with PC. Previous studies have shown that, at least for simple datasets, the number of inference steps can remain close to the depth without sacrificing performance (Pinchetti et al., 2024; Innocenti et al., 2025), despite no longer reaching an equilibrium. PC inference can also be accelerated using BP (Goemaere et al., 2025) and other activity initialisations (Pinchetti et al., 2026).

Setting aside the increased memory cost of most of these methods, we speculate that significant speed-ups for PC inference—and so potential compute savings over BP—are more likely to come from analog hardware implementations that physically realise the network dynamics (Momeni et al., 2025; Aifer et al., 2025; Montanari et al., 2026; Wright et al., 2026). Whether PC can be successfully implemented on such hardware remains an important open question.

**Biologically plausible attention.** One important limitation of our transformer experiment (Figure 4)—with potential implications for analog hardware (Bacvanski et al., 2025)—is that the self-attention mechanism (Vaswani et al., 2017) is non-local, in that it requires computing pairwise interactions between every token across the input sequence. It could be interesting to investigate attention mechanisms where the softmax itself is the result of gradient descending an energy function (Singh & Buckley, 2023). Such mechanisms are closely related to modern Hopfield networks (Krotov & Hopfield, 2016; Ramsauer et al., 2020; Hoover et al., 2023) and have some biologically plausible implementations (Kozachkov et al., 2025; Kafraj et al., 2026).

**Training to convergence.** All our experiments trained models for only 100 steps. This was partly because our aim was to test the theory and partly due to the high computational cost of PC inference, which for the deepest models required up to 100k steps to converge (see §A.8). It could be interesting, as a proof of concept, to show that PC can train a nano-GPT-style model to convergence with competitive BP performance. This experiment would also allow one to leverage the hyperparameter transfer property of $\mu$P (Yang & Hu, 2021), which we did not validate here but predict would hold in the regime where PC converges to BP.

---

[3]We thank one of the reviewers for this suggestion.

## Acknowledgements

FI and RB acknowledge funding from the Wellcome Trust grant 313955/Z/24/Z. RB acknowledges funding from the Medical Research Council grants UKRI/MR/B000936/1 and MC_UU_00003/1. EMA acknowledges funding by UM6P.

## Impact Statement

This paper presents work whose goal is to advance the field of Machine Learning. There are many potential societal consequences of our work, none which we feel must be specifically highlighted here.

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

# A. Appendix

## Table of Contents

### A.1. General notation

Vectors $\mathbf{v}$ and matrices $\mathbf{A}$ are typed in bold, while scalars $c$ and $C$ are in non-bold, including element-wise indexing of vectors $v_i$ and matrices $A_{ij}$. All vectors are column-oriented unless otherwise stated (e.g. the output weights $\mathbf{w}^{(L)} \in \mathbb{R}^{1 \times N}$), and the dot product between two vectors is denoted as $\mathbf{u} \cdot \mathbf{v}$. $\| \cdot \|$ is the $l_2$ or Euclidean norm, $\| \cdot \|_F$ is the Frobenius norm, and $\text{vec}(\cdot)$ is the row-vector operator. We will denote expectations with respect to model parameters $\boldsymbol{\theta}$ with angle brackets: $\langle \cdot \rangle := \mathbb{E}_{\boldsymbol{\theta}}[\cdot]$. $\mathbf{I}_N$ is the identity matrix of size $N$, and $x = \Theta_N(1)$ indicates that $x$ neither vanishes nor blows up with $N$, i.e. $0 < \liminf_{N \to \infty} |x(N)| \leq \limsup_{N \to \infty} |x(N)| < \infty$.

### A.2. Related work

**Scaling limits of neural networks.** There is a long history of studying neural networks by taking infinite limits of several coarse (thermodynamic) variables (Amit et al., 1987), such as the number of data samples or the size of the model. Limits taken with respect to model size are the most relevant for our purposes and have also had great practical impact.[4]

---

[4]"Scaling laws", where model size, training time and dataset size are all scaled, have arguably been the most impactful (Kaplan et al., 2020; Hoffmann et al., 2022). However, in contrast to the theoretical limits reviewed above, scaling laws originated as empirical observations.

**Early results on the infinite-width limit.** The correspondence between Gaussian Processes and the function computed by feedforward networks at initialisation (NNGP) was one of the first results on the infinite-width limit (Neal, 1996; Lee et al., 2017; Matthews et al., 2018). Jacot et al. (2018) then famously showed that the gradient descent training dynamics of wide networks are equivalent to a kernel method termed the "neural tangent kernel" (NTK). However, in this regime, the parameters stay close to their initialisation, and the network features or preactivations barely move at large width, resulting in "lazy" learning (Chizat et al., 2019; Lee et al., 2019).

**The mean-field/$\mu$P parameterisation.** Motivated by the lack of feature learning in the NTK regime, Yang & Hu (2020) derived a parameterisation in the infinite-width limit where the feature updates are "maximal" in a well-defined sense (see Desideratum 3 in §2.3)—hence the "maximal update parameterisation" or $\mu$P. Moreover, Yang & Hu (2021) showed that under $\mu$P, hyperparameters such as the learning rate also remain stable across model widths, enabling zero-shot hyperparameter transfer. While Yang & Hu (2020) derived the theoretical limit using the "Tensor Programs" framework, Bordelon & Pehlevan (2022b) later described the same limit using a dynamical mean field theory inspired from statistical physics. These and other parameterisations are explained in more detail in §3 and precisely defined in Table 2.

**Depth-aware parameterisations.** $\mu$P has been extended to a variety of architectures and optimisers. The extensions to infinitely deep residual networks (Yang & Hu, 2020; Bordelon et al., 2023; Dey et al., 2025), for which the order of the width and depth limits does not matter (i.e. it commutes) (Hayou & Yang, 2023), are most relevant for our purposes. As reviewed in §4, such parameterisations mainly introduce a depth-dependent rescaling $L^{-\alpha}$ of the residual branches and enable hyperparameter transfer across model depths.

**Reparameterisations of PCNs.** Our work is most closely related to Ishikawa et al. (2024) and Innocenti et al. (2025). Ishikawa et al. (2024) derived an infinite-width, $\mu$P-like parameterisation for linear PCNs. The key difference with our work is that Ishikawa et al. (2024) showed the *existence* of a parameterisation where linear PCNs implement BP with GD (see their Corollary 4.3). By contrast, we prove a much stronger result: if one would like to satisfy the $\mu$P desiderata, then *necessarily*, $\mu$P is the only stable and feature-learning parameterisation for PC (Theorems 1-2). Our work also differs from Ishikawa et al. (2024) in the following respects: (i) we derive a slightly more general set of stable and feature-learning parameterisations, in the same way as Bordelon & Pehlevan (2022b) recovered $\mu$P (Yang & Hu, 2021); (ii) we extend the parameterisation to depth; and (iii) we do not include scalings of the layer PC energies or activity learning rates (Eq. 2).

Innocenti et al. (2025), on the other hand, used a parameterisation inspired by depth extensions of $\mu$P (with $\alpha = 1/2$) (Yang et al., 2023; Bordelon et al., 2023) to stabilise the forward pass of deep PCNs ("$\mu$PC"). Notably, they found that this parameterisation enabled the training of 100+ layer residual networks on standard classification benchmarks with hyperparameter transfer, at competitive performance with BP. However, as noted in §4, beyond the forward pass stability, it remains unclear whether this parameterisation satisfies the depth feature-learning desideratum (5) for PC. We clearly answer this question in the affirmative in §4 with Theorem 2.

It is also interesting to note that Innocenti et al. (2025) used Adam for their experiments but did not rescale the learning rate according to the theoretical prescriptions (see §A.6), noting that such scalings led to unstable training. This could be due to the fact that they failed to reach convergence of the PC inference dynamics under this parameterisation (see §5). Finally, Innocenti et al. (2025) also proved a convergence result to BP similar to our Corollary 4.2, which we discuss below.

Finally, Goemaere et al. (2025) proposed a reparameterisation of PCNs where errors, instead of activities, are optimised with BP during inference ("ePC"). Since this reparameterisation changes the network's forward pass and the standard PC energy (Eq. 2), it remains unclear whether ePC is stable and rich with respect to both width and depth—and so whether it is fully scalable. This could be determined by extending our analysis to the ePC energy.

**Previous correspondences between PC and BP.** Early research on PC as a learning algorithm established two main conditions under which PC can approximate or compute exactly the same gradients as BP. First, as also briefly reviewed in §3, PC approximates BP when the output energy is much larger than the other layer energies (Whittington & Bogacz, 2017), which occurs when the equilibrium of the activities is close to the forward pass. Second, if the weight updates are coordinated with specific inference steps, then PC exactly implements BP (Song et al., 2020). Various extensions and generalisations of these results have been formulated (Millidge et al., 2022b; Rosenbaum, 2022; Salvatori et al., 2021; Millidge et al., 2022a).

More recently, Innocenti et al. (2025) proved convergence, *at initialisation*, of the equilibrated energy to the MSE loss under

the same depth parameterisation of linear residual networks considered in §4. Our Corollary 4.2 shows that this result holds throughout training. We also note that, unlike our results, Innocenti et al. (2025) found that the empirical correspondence between the equilibrated energy and the MSE loss deteriorated with training at large width and depth. This could have been due to numerical instabilities in their computation of the equilibrated energy, which was based on inverting a highly ill-conditioned Hessian matrix.

### A.3. Table of width-aware parameterisations for BP

The Table below shows the scalings defining the width-aware parameterisations reviewed in §2.3.

*Table 2.* Summary of $(a_\ell, b_\ell, c, d)$ parameterisations derived for BP at large width. For a description of the width-scaling exponents, see Table 1. The standard parameterisation (SP) includes all common frameworks (e.g. PyTorch) that initialise weights with variance inversely proportional to their input dimension. The NTK parameterisation is from Jacot et al. (2018), while the mean-field and $\mu$P parameterisations refer to those introduced in Bordelon & Pehlevan (2022b) and Yang & Hu (2021), respectively. Note that we omit any data-dependent normalisation (e.g. $1/\sqrt{D}$) of the first layer's preactivations (see Eq. 10).

| Exponent | | SP | NTK | Mean-field | $\mu$P |
|---|---|---|---|---|---|
| $a_1$ | | 0 | 0 | 0 | $-1/2$ |
| $b_1$ | | 0 | 0 | 0 | 1 |
| $a_\ell,$ | $\ell \in \{2, \ldots, L\}$ | 0 | 1/2 | 1/2 | 0 |
| $b_\ell,$ | $\ell \in \{2, \ldots, L\}$ | 1 | 0 | 0 | 1 |
| $c$ | | 0 | 0 | 0 | 1 |
| $d$ | | 0 | 0 | 1/2 | 1/2 |

### A.4. Derivation of width scalings for linear PCNs

Based on the the general $(a_\ell, b_\ell, c, d)$ parameterisation of linear MLPs defined in §2.2 and the equilibrated energy of linear PCNs (Eq. 5), this section derives the with scaling exponents for PC that satisfy the 3 Desiderata listed in §2.3, proving Theorem 1. For convenience, we recall our MLP parameterisation:

$$f_\mu = \frac{1}{\gamma} h_\mu^{(L)}, \tag{20}$$

$$h_\mu^{(L)} = \frac{1}{N^{a_L}} \mathbf{w}^{(L)} \mathbf{h}_\mu^{(L-1)}, \qquad\qquad w_i^{(L)} \sim \mathcal{N}\left(0, \frac{1}{N^{b_L}}\right), \tag{21}$$

$$\mathbf{h}_\mu^{(\ell)} = \frac{1}{N^{a_\ell}} \mathbf{W}^{(\ell)} \mathbf{h}_\mu^{(\ell-1)}, \qquad\qquad W_{ij}^{(\ell)} \sim \mathcal{N}\left(0, \frac{1}{N^{b_\ell}}\right), \tag{22}$$

$$\mathbf{h}_\mu^{(1)} = \frac{1}{N^{a_1}\sqrt{D}} \mathbf{W}^{(1)} \mathbf{x}_\mu, \qquad\qquad W_{ij}^{(1)} \sim \mathcal{N}\left(0, \frac{1}{N^{b_1}}\right), \tag{23}$$

where again recall that $\gamma = \gamma_0 N^d$. Under this parameterisation, the equilibrated PC energy (Eq. 5) is given by

$$\mathcal{F}^*(\boldsymbol{\theta}) = \frac{1}{s(\boldsymbol{\theta})} \mathcal{L}(\boldsymbol{\theta}), \tag{24}$$

$$s(\boldsymbol{\theta}) = 1 + \gamma^{-2} \sum_{\ell=2}^{L} N^{-2\sum_{k=\ell}^{L} a_k} \|\mathbf{W}^{(L:\ell)}\|^2. \tag{25}$$

For what follows, it will also be useful to define the gradient flow on the equilibrated energy (Eq. 5)

$$\frac{d\boldsymbol{\theta}}{dt} = -\eta \frac{\partial \mathcal{F}^*}{\partial \boldsymbol{\theta}} = -\eta \left( \frac{1}{s(\boldsymbol{\theta})} \frac{\partial \mathcal{L}}{\partial \boldsymbol{\theta}} - \frac{\mathcal{L}}{s(\boldsymbol{\theta})^2} \frac{\partial s(\boldsymbol{\theta})}{\partial \boldsymbol{\theta}} \right), \tag{26}$$

which decomposes into two terms: (i) a rescaled loss gradient, and (ii) an extra term depending on the parameter derivative of the rescaling. These terms will appear in the analysis of the second and third desiderata (§A.4.3-A.4.4). Also recall that $\eta = \eta_0 \gamma^2 N^{-c}$. The following subsections derive specific constraints for PC for each desideratum listed in §2.3. Our notation and derivation style closely follow those of Bordelon & Pehlevan (2022b), with a physics-level rigour of analysis.

A.4.1. DESIDERATUM 1: PREACTIVATIONS ARE $\Theta_N(1)$ AT INITIALISATION

Verification of this desideratum does not differ between BP and PC, since the forward pass is algorithm-independent. We emphasise that the following derivation is not novel, and similar calculations can in fact be traced to classic weight initialisation schemes (LeCun et al., 2002; He et al., 2015). However, we include it for completeness.

Starting from the activations of the first layer, note that $\mathbf{h}_\mu^{(1)}$ is a sum of Gaussian random variables (Eq. 10). Its statistics are therefore fully specified by its mean and covariance:

$$\left\langle h_{\mu,i}^{(1)} \right\rangle = \frac{1}{N^{a_1}\sqrt{D}} \sum_k^D \left\langle W_{ik}^{(1)} \right\rangle x_{\mu,k} = 0, \tag{27}$$

$$\left\langle h_{\mu,i}^{(1)} h_{\nu,j}^{(1)} \right\rangle = \frac{1}{N^{2a_1}D} \sum_k^D \sum_{k'}^D \left\langle W_{ik}^{(1)}(0) W_{jk'}^{(1)}(0) \right\rangle x_{\mu,k} x_{\nu,k'} \tag{28}$$

$$= \frac{1}{N^{2a_1}D} \sum_k^D \sum_{k'}^D \delta_{ij}\delta_{kk'} \frac{1}{N^{b_1}} x_{\mu,k} x_{\nu,k'} \tag{29}$$

$$= \delta_{ij} \frac{1}{N^{2a_1+b_1}D} \sum_k^D x_{\mu,k} x_{\nu,k}, \tag{30}$$

where $W_{ij}^{(1)}(0)$ denotes the value of a weight at initialisation ($t = 0$). To keep the covariance $\Theta_N(1)$ as $N \to \infty$, we therefore need the constraint $2a_1 + b_1 = 0$. Moving on to the next layer, its mean and covariance are given by

$$\left\langle h_{\mu,i}^{(2)} \right\rangle = \frac{1}{N^{a_\ell}} \sum_k^N \left\langle W_{ik}^{(\ell)}(0) \right\rangle \left\langle h_{\mu,k}^{(1)} \right\rangle = 0, \tag{31}$$

$$\left\langle h_{\mu,i}^{(2)} h_{\nu,j}^{(2)} \right\rangle = \frac{1}{N^{2a_\ell}} \sum_k^N \sum_{k'}^N \left\langle W_{ik}^{(\ell)}(0) W_{jk'}^{(\ell)}(0) \right\rangle \left\langle h_{\mu,k}^{(1)} h_{\nu,k'}^{(1)} \right\rangle \tag{32}$$

$$= \frac{1}{N^{2a_\ell}} \sum_k^N \sum_{k'}^N \delta_{ij}\delta_{kk'} \frac{1}{N^{b_\ell}} \left\langle h_{\mu,k}^{(1)} h_{\nu,k'}^{(1)} \right\rangle \tag{33}$$

$$= \delta_{ij} \frac{1}{N^{2a_\ell+b_\ell}} \sum_k^N \left\langle h_{\mu,k}^{(1)} h_{\nu,k}^{(1)} \right\rangle \tag{34}$$

$$= \delta_{ij} \frac{1}{N^{2a_\ell+b_\ell-1}} \left\langle h_{\mu,1}^{(1)} h_{\nu,1}^{(1)} \right\rangle, \tag{35}$$

where in the last step we use the fact that neurons $h_{\mu,k}^{(1)}$ are i.i.d. across $k$, and so the average over $k$ equals the value for any fixed $k$, such as $k = 1$. This argument can be generalised inductively to deeper layers, and so overall, we obtain the following constraints for Desideratum 1

$$\boxed{\begin{aligned} 2a_1 + b_1 &= 0, \\ 2a_\ell + b_\ell &= 1. \end{aligned}} \tag{36}$$

Note that the constraint for the first layer differs from that of other layers because it is the only layer whose input dimension does not grow to infinity (as the dataset is assumed to be constant with respect to the width). We will find an analogous result for the next desideratum. Also note that, to satisfy this desideratum, one can choose to scale the initialisation variance of the weights (as in the SP) or the preactivations themselves (as in the NTK and $\mu$P parameterisations), or both if one wishes.

A.4.2. USEFUL ASYMPTOTIC RESULTS FOR MLPS

It is useful, at this point, to present a number of asymptotic results that will simplify the analysis of the subsequent desiderata. These results concern the following objects:

- "feature and gradient kernels" (defined below);

- the derivative of the network function with respect to the weights of any layer, $\partial f / \partial \mathbf{W}^{(\ell)}$;

- the equilibrated energy rescaling $s(\boldsymbol{\theta})$; and

- the derivative of the equilibrated energy rescaling with respect to layer weights, $\partial s / \partial \mathbf{W}^{(\ell)}$.

Note that the first two objects have been studied for BP, while the last two are specific to PC. In §A.5.2, we will extend the last two results to linear residual networks.

**Feature and gradient kernels.** First, following Bordelon & Pehlevan (2022b) we define the "feature kernels"

$$\Phi_{\mu\nu}^{(\ell)} := \frac{1}{N} \sum_{k=1}^{N} h_{\mu,k}^{(\ell)} h_{\nu,k}^{(\ell)} = \frac{1}{N} \mathbf{h}_\mu^{(\ell)} \cdot \mathbf{h}_\nu^{(\ell)}, \quad \ell \in \{1, \ldots, L-1\}, \tag{37}$$

$$\Phi_{\mu\nu}^{(0)} := \frac{1}{D} \mathbf{x}_\mu \cdot \mathbf{x}_\nu, \tag{38}$$

as well as the "gradient kernels"

$$G_{\mu\nu}^{(\ell)} := \frac{1}{N} \sum_{k=1}^{N} g_{\mu,k}^{(\ell)} g_{\nu,k}^{(\ell)} = \frac{1}{N} \mathbf{g}_\mu^{(\ell)} \cdot \mathbf{g}_\nu^{(\ell)}, \quad \mathbf{g}_\mu^{(\ell)} = \sqrt{N} \frac{\partial h_\mu^{(L)}}{\partial \mathbf{h}_\mu^{(\ell)}}, \tag{39}$$

$$G_{\mu\nu}^{(L)} = 1. \tag{40}$$

These objects (and their nonlinear versions) have been previously studied in the signal propagation literature (Poole et al., 2016; Schoenholz et al., 2016) under the names of forward and backward kernels, respectively. As shown by Bordelon & Pehlevan (2022b), under the considered parameterisation, both the forward and backward kernels concentrate around their expectation as $N \to \infty$ and are both width-stable:

$$\left\langle \Phi_{\mu\nu}^{(\ell)} \right\rangle = \frac{1}{N} \sum_{k=1}^{N} \left\langle h_{\mu,k}^{(\ell)} h_{\nu,k}^{(\ell)} \right\rangle = \Theta_N(1), \tag{41}$$

$$\left\langle G_{\mu\nu}^{(\ell)} \right\rangle = \frac{1}{N} \sum_{k=1}^{N} \left\langle g_{\mu,k}^{(\ell)} g_{\nu,k}^{(\ell)} \right\rangle = \Theta_N(1). \tag{42}$$

**Derivative of the network function with respect to layer weights, $\partial f / \partial \mathbf{W}^{(\ell)}$.** It will also be useful to characterise the order of the parameter derivative of the network function with respect to $N$

$$\frac{\partial f_\mu}{\partial W_{ij}^{(\ell)}} = \frac{1}{\gamma} \frac{\partial h_\mu^{(L)}}{\partial W_{ij}^{(\ell)}} = \frac{1}{\gamma \sqrt{N}} \underbrace{\left( \sqrt{N} \frac{\partial h_\mu^{(L)}}{\partial h_{\mu,i}^{(\ell)}} \right)}_{g_i^{(\ell)}} \frac{\partial h_{\mu,i}^{(\ell)}}{\partial W_{ij}^{(\ell)}} \tag{43}$$

$$= \gamma_0^{-1} N^{-d-1/2} g_i^{(\ell)} \frac{1}{N^{a_\ell}} h_{\mu,j}^{(\ell-1)} = \Theta(N^{-d-a_\ell-1/2}). \tag{44}$$

**Equilibrated energy rescaling $s(\boldsymbol{\theta})$ (Eq. 25).** For many of the subsequent analyses, it will be useful to understand the asymptotic behaviour of the equilibrated energy rescaling under our parameterisation, with both the network width $N$ and depth $L$. As shown in Ishikawa et al. (2024), by the law of large numbers, the squared norm term in the equilibrated energy rescaling (Eq. 25) also converges in the infinite-width limit, giving

$$\left\langle \|\mathbf{W}^{(L:\ell)}\|^2 \right\rangle = \left\langle \|\mathbf{w}^{(L)} \mathbf{W}^{(L-1)} \ldots \mathbf{W}^{(\ell)}\|^2 \right\rangle = N^{1-b_L} \prod_{k=\ell}^{L-1} N^{1-b_k} = N^{\sum_{k=\ell}^{L}(1-b_k)}. \tag{45}$$

For intuition, this is easier to see for the one-hidden-layer case where $\left\langle \|\mathbf{w}^{(L)}\|^2 \right\rangle = \left\langle \sum_k^N (w_k^{(L)})^2 \right\rangle = N^{1-b_L}$. If we plug in the forward pass constraint $b_\ell = 1 - 2a_\ell$ from the first desideratum (§A.4.1), Eq. 45 simplifies to $N^{\sum_{k=\ell}^{L}(1-b_k)} = N^{2\sum_{k=\ell}^{L} a_k}$.

This leads to the useful result

$$\sum_{\ell=2}^{L} N^{-2\sum_{k=\ell}^{L} a_k} \|\mathbf{W}^{(L:\ell)}\|^2 = \sum_{\ell=2}^{L} N^{-2\sum_{k=\ell}^{L} a_k} N^{2\sum_{k=\ell}^{L} a_k} = \Theta_N(1), \tag{46}$$

which we will rely on in many subsequent derivations. We emphasise that this result assumes the forward pass constraint from the first desideratum (which is what leads to the cancellation of the scaled norm terms). For the later depth-scaling analysis (§A.5), it will be important to realise that, if the depth is not constant with respect to the width, then Eq. 46 becomes $\Theta(L)$. If one assumes growing width *and* depth, we therefore get the following result:

$$s(\boldsymbol{\theta}) = 1 + \Theta(\gamma^{-2}L) = 1 + \Theta(L/N^{2d}), \quad N, L \to \infty. \tag{47}$$

**Derivative of the equilibrated energy rescaling with respect to layer weights, $\partial s/\partial \mathbf{W}^{(\ell)}$.** For the analysis of the gradient flow on the equilibrated energy (Eq. 26), it will also be useful to characterise the asymptotic behaviour of the *derivative* of the rescaling with respect to any weight matrix for $\ell > 1$:

$$\frac{\partial s(\boldsymbol{\theta})}{\partial \mathbf{W}^{(\ell)}} = 2\gamma^{-2} \sum_{k=2}^{\ell} N^{-2\sum_{j=k}^{L} a_j} \left( \mathbf{W}^{(L:\ell+1)\top} \mathbf{W}^{(L:k)} \mathbf{W}^{(\ell-1:k)\top} \right), \tag{48}$$

with boundary cases $\mathbf{W}^{(L:L+1)} = \mathbf{W}^{(\ell-1:\ell)} = \mathbf{I}$. Let us first look at the structure of the matrix product. Note that $\mathbf{W}^{(L:\ell+1)\top} \in \mathbb{R}^{N \times 1}$, $\mathbf{W}^{(L:k)} \in \mathbb{R}^{1 \times N}$, and $\mathbf{W}^{(\ell-1:k)\top} \in \mathbb{R}^{N \times N}$. Given that any single weight $W_{ij}^{(\ell)}$ has magnitude $\Theta(N^{-b_\ell/2}) = \Theta(N^{a_\ell - 1/2})$ under the forward pass constraint, we have

$$W_{i1}^{(L:\ell+1)\top} = \Theta \left( N^{\sum_{m=\ell+1}^{L} a_m - 1/2} \right), \tag{49}$$

$$W_{1j}^{(L:k)} = \Theta \left( N^{\sum_{m=k}^{L} a_m - 1/2} \right), \tag{50}$$

$$W_{ij}^{(\ell-1:k)\top} = \Theta \left( N^{\sum_{m=k}^{\ell-1} a_m - 1/2} \right). \tag{51}$$

Combining the exponents, we get

$$\left( \sum_{m=\ell+1}^{L} a_m - 1/2 \right) + \left( \sum_{m=k}^{L} a_m - 1/2 \right) + \left( \sum_{m=k}^{\ell-1} a_m - 1/2 \right) = 2 \sum_{m=k}^{L} a_m - a_\ell - 3/2. \tag{52}$$

Now, for convenience let the matrix product be denoted by $\mathbf{M} = \mathbf{a}\mathbf{b}^\top \mathbf{C}$, where $\mathbf{a} = \mathbf{W}^{(L:\ell+1)\top}$, $\mathbf{b}^\top = W_{1j}^{(L:k)}$, and $\mathbf{C} = \mathbf{W}^{(\ell-1:k)\top}$. An element of this product is given by

$$M_{ij} = \sum_{k=1}^{N} a_i b_{1k}^\top C_{kj}, \tag{53}$$

where we note the summation over $N$. Combining this with the scaling of the matrix product (Eq. 52), each entry of the derivative (Eq. 48) scales as

$$\frac{\partial s}{\partial W_{ij}^{(\ell)}} = 2\gamma^{-2} \sum_{k=2}^{\ell} N^{-2\sum_{j=k}^{L} a_j} \left( N^{1-3/2} \cdot N^{2\sum_{j=k}^{L} a_j - a_\ell} \right) \tag{54}$$

$$= \Theta(LN^{-2d-a_\ell-1/2}), \quad N, L \to \infty. \tag{55}$$

A.4.3. DESIDERATUM 2: NETWORK PREDICTIONS EVOLVE IN $\Theta_N(1)$ TIME DURING TRAINING

This desideratum requires that $df/dt = \Theta_N(1)$. Let us unpack this:

$$\frac{df(\mathbf{x}_\mu; \boldsymbol{\theta})}{dt} = \frac{\partial f_\mu}{\partial \boldsymbol{\theta}} \cdot \frac{d\boldsymbol{\theta}}{dt} \tag{56}$$

$$= -\eta \left[ \frac{1}{s(\boldsymbol{\theta})P} \sum_{\nu=1}^{P} \frac{\partial f_\mu}{\partial \boldsymbol{\theta}} \cdot \frac{\partial \mathcal{L}_\nu}{\partial \boldsymbol{\theta}} - \frac{\mathcal{L}_\nu}{s(\boldsymbol{\theta})^2} \frac{\partial f_\mu}{\partial \boldsymbol{\theta}} \cdot \frac{\partial s(\boldsymbol{\theta})}{\partial \boldsymbol{\theta}} \right] \tag{57}$$

$$= \underbrace{\eta \frac{1}{s(\boldsymbol{\theta})P} \sum_{\nu=1}^{P} \frac{\partial f_\mu}{\partial \boldsymbol{\theta}} \cdot \frac{\partial f_\nu}{\partial \boldsymbol{\theta}} \Delta_\nu}_{q} + \underbrace{\eta \frac{\mathcal{L}_\nu}{s(\boldsymbol{\theta})^2} \frac{\partial f_\mu}{\partial \boldsymbol{\theta}} \cdot \frac{\partial s(\boldsymbol{\theta})}{\partial \boldsymbol{\theta}}}_{r}, \tag{58}$$

where $\Delta_\nu = -\partial \mathcal{L}_\nu / \partial f_\nu$. Note that, in contrast to BP, the evolution of the network function for PC is given by two terms: (i) a rescaled NTK term $q$, where $\partial f_\mu / \partial \boldsymbol{\theta} \cdot \partial f_\nu / \partial \boldsymbol{\theta}$ is the so-called NTK; and (ii) an extra term $r$ coming from the gradient of the equilibrated energy (see Eq. 26). Below, we analyse the order of these two terms separately, before combining them. Note that we will make use of the asymptotic results derived in §A.4.2.

**Rescaled NTK term $q$.** Let us first recall from our parameterisation that $\eta = \eta_0 \gamma^2 N^{-c}$. Given that the dataset size and loss delta $\Delta_\nu$ are constants with respect to the network width, the rescaled NTK term $q$ (in Eq. 58) requires the following to be $\Theta_N(1)$:

$$\frac{\gamma^2}{N^c} \frac{1}{s(\boldsymbol{\theta})} \frac{\partial f_\mu}{\partial \boldsymbol{\theta}} \cdot \frac{\partial f_\nu}{\partial \boldsymbol{\theta}} = \frac{1}{N^c s(\boldsymbol{\theta})} \frac{\partial h_\mu^{(L)}}{\partial \boldsymbol{\theta}} \cdot \frac{\partial h_\nu^{(L)}}{\partial \boldsymbol{\theta}} \tag{59}$$

$$= \frac{1}{N^c s(\boldsymbol{\theta})} \sum_{\ell=1}^{L} \sum_{i,j} \frac{\partial h_\mu^{(L)}}{\partial W_{ij}^{(\ell)}} \frac{\partial h_\nu^{(L)}}{\partial W_{ij}^{(\ell)}} \tag{60}$$

$$= \frac{1}{N^c s(\boldsymbol{\theta})} \sum_{\ell=1}^{L} \sum_{i,j} \sum_{m,n} \frac{\partial h_\mu^{(L)}}{\partial h_{\mu,m}^{(\ell)}} \frac{\partial h_{\mu,m}^{(\ell)}}{\partial W_{ij}^{(\ell)}} \frac{\partial h_\nu^{(L)}}{\partial h_{\nu,n}^{(\ell)}} \frac{\partial h_{\nu,n}^{(\ell)}}{\partial W_{ij}^{(\ell)}} \tag{61}$$

$$= \frac{1}{N^c s(\boldsymbol{\theta})} \left[ \frac{1}{N^{2a_L}} \mathbf{h}_\mu^{(L-1)} \cdot \mathbf{h}_\nu^{(L-1)} + \sum_{\ell=2}^{L-1} \frac{\partial h_\mu^{(L)}}{\partial \mathbf{h}_\mu^{(\ell)}} \cdot \frac{\partial h_\nu^{(L)}}{\partial \mathbf{h}_\nu^{(\ell)}} \frac{1}{N^{2a_\ell}} \mathbf{h}_\mu^{(\ell-1)} \cdot \mathbf{h}_\nu^{(\ell-1)} \right. \tag{62}$$

$$\left. + \frac{\partial h_\mu^{(L)}}{\partial \mathbf{h}_\mu^{(1)}} \cdot \frac{\partial h_\nu^{(L)}}{\partial \mathbf{h}_\nu^{(1)}} \frac{1}{N^{2a_1} D} \mathbf{x}_\mu \cdot \mathbf{x}_\nu \right] \tag{63}$$

$$= \frac{1}{N^c s(\boldsymbol{\theta})} \left[ \frac{1}{N^{2a_L - 1}} \Phi_{\mu\nu}^{(L-1)} + \sum_{\ell=2}^{L-1} \frac{1}{N^{2a_\ell - 1}} G_{\mu\nu}^{(\ell)} \Phi_{\mu\nu}^{(\ell-1)} + \frac{1}{N^{2a_1}} G_{\mu\nu}^{(1)} \Phi_{\mu\nu}^{(0)} \right]. \tag{64}$$

Again, note that this is simply the NTK scaled by the learning rate and the equilibrated energy rescaling. As mentioned in §A.4.2, the feature and gradient kernels concentrate in the limit and are $\Theta_N(1)$. Similarly, we showed that the equilibrated energy rescaling also converges, scaling as $s(\boldsymbol{\theta}) = 1 + \Theta(N^{-2d})$ at constant depth. Using these facts, we get

$$\frac{1}{N^c(1 + N^{-2d})} \left[ \frac{1}{N^{2a_L - 1}} \Theta_N(1) + \sum_{\ell=2}^{L-1} \frac{1}{N^{2a_\ell - 1}} \Theta_N(1) \Theta_N(1) + \frac{1}{N^{2a_1}} \Theta_N(1) \Theta_N(1) \right]. \tag{65}$$

We see that the denominator contains two competing scaling factors: $N^c$ and $N^{c-2d}$. As $N \to \infty$, the behavior of the network is dictated by whichever of these terms is dominant. We require the dominant exponent to be zero, which implies

$$\max(c + 2a_\ell - 1, \ c + 2a_\ell - 1 - 2d) = 0 \implies \boxed{c + 2a_\ell - 1 = \min(0, 2d)}, \tag{66}$$

$$\max(c + 2a_1, \ c + 2a_1 - 2d) = 0 \implies \boxed{c + 2a_1 = \min(0, 2d)}. \tag{67}$$

Note that, similar to the first desideratum (§A.4.1), we find a different constraint for the first layer, again because it is the only layer whose input dimension is not assumed to grow to infinity.

**Extra energy gradient term $r$.** Having analysed the first term, $q$, controlling the network function's evolution (Eq. 58), we now examine the second term, $r$, which is specific to PC. Given that the loss is $\Theta_N(1)$ under the previous forward pass constraint, the gradient term $r$ requires the following to be $\Theta_N(1)$:

$$\frac{\gamma^2}{N^c}\frac{1}{s(\boldsymbol{\theta})^2}\frac{\partial f_\mu}{\partial \boldsymbol{\theta}} \cdot \frac{\partial s(\boldsymbol{\theta})}{\partial \boldsymbol{\theta}} = \frac{\gamma^2}{N^c}\frac{1}{s(\boldsymbol{\theta})^2}\sum_{\ell=2}^{L}\left\langle \frac{\partial f_\mu}{\partial \mathbf{W}^{(\ell)}}, \frac{\partial s}{\partial \mathbf{W}^{(\ell)}}\right\rangle_F = \frac{\gamma^2}{N^c}\frac{1}{s(\boldsymbol{\theta})^2}\sum_{\ell=2}^{L}\sum_{i,j}^{N}\frac{\partial f_\mu}{\partial W_{ij}^{(\ell)}}\cdot\frac{\partial s(\boldsymbol{\theta})}{\partial W_{ij}^{(\ell)}} \tag{68}$$

where $\langle \mathbf{A}, \mathbf{B}\rangle_F$ denotes the Frobenius inner product. From the results of §A.4.2 (specifically Eqs. 44 & 55), we know how the entries of the terms of the inner product scale with $N$, giving

$$\frac{\gamma^2}{N^c}\frac{1}{s(\boldsymbol{\theta})^2}\sum_{\ell=2}^{L}\sum_{i,j}^{N}N^{-d-a_\ell-1/2}\cdot LN^{-2d-a_\ell-1/2} \tag{69}$$

$$= \frac{\gamma^2}{N^c}\frac{1}{s(\boldsymbol{\theta})^2}LN^2LN^{-3d-2a_\ell-1} \tag{70}$$

$$= \frac{\gamma_0^2}{N^{c-2d}(1+N^{-2d})^2}L^2N^{1-3d-2a_\ell} \tag{71}$$

$$= \frac{\gamma_0^2 L^2}{N^{c+d+2a_\ell-1}(1+N^{-2d})^2}. \tag{72}$$

As before, we look for the largest exponent, obtaining

$$\boxed{c+d+2a_\ell-1 = \min(0, 2d)}, \tag{73}$$

So, the scaling of the extra energy gradient term $r$ depends on the output scaling exponent $d$ (Eq. 11). In the next section, we will see that the constraint of the last desideratum ($d = 1/2$) forces this term to vanish.

### A.4.4. DESIDERATUM 3: FEATURES EVOLVE IN $\Theta_N(1)$ TIME DURING TRAINING

This final desideratum requires that $dh_i^{(\ell)}/dt = \Theta_N(1)$. Starting again with the first layer, it is useful to write the learning dynamics of the first weight matrix

$$\frac{d\mathbf{W}^{(1)}}{dt} = -\eta\frac{\partial \mathcal{F}_\nu^*}{\partial \mathbf{W}^{(1)}} = -\eta\frac{1}{s(\boldsymbol{\theta})P}\sum_{\nu=1}^{P}\frac{\partial \mathcal{L}_\nu}{\partial \mathbf{W}^{(1)}} \tag{74}$$

$$= \eta\frac{1}{s(\boldsymbol{\theta})P}\sum_{\nu=1}^{P}\Delta_\nu\frac{\partial f_\nu}{\partial \mathbf{W}^{(1)}} \tag{75}$$

$$= \frac{\eta_0\gamma^2}{N^c s(\boldsymbol{\theta})}\frac{1}{P}\sum_{\nu=1}^{P}\frac{\Delta_\nu}{\gamma}\frac{\partial h_\nu^{(L)}}{\partial \mathbf{W}^{(1)}} \tag{76}$$

$$= \frac{\eta_0\gamma}{N^c s(\boldsymbol{\theta})}\frac{1}{P}\sum_{\nu=1}^{P}\Delta_\nu\frac{\partial h_\nu^{(L)}}{\partial \mathbf{h}_\nu^{(1)}}\frac{\partial \mathbf{h}_\nu^{(1)}}{\partial \mathbf{W}^{(1)}} \tag{77}$$

$$= \frac{\eta_0\gamma_0}{N^{c+a_1-d}s(\boldsymbol{\theta})}\frac{1}{\sqrt{D}P}\sum_{\nu=1}^{P}\frac{\Delta_\nu\sqrt{N}}{\sqrt{N}}\frac{\partial h_\nu^{(L)}}{\partial \mathbf{h}_\nu^{(1)}}\mathbf{x}_\nu^\top \tag{78}$$

$$= \frac{\eta_0\gamma_0}{N^{c+a_1-d+1/2}s(\boldsymbol{\theta})}\frac{1}{\sqrt{D}P}\sum_{\nu=1}^{P}\Delta_\nu\mathbf{g}_\nu^{(1)}\mathbf{x}_\nu^\top, \tag{79}$$

where recall that $\Delta_\nu = -\partial \mathcal{L}_\nu/\partial f_\nu$. Since the equilibrated energy rescaling (Eq. 25) does not depend on the weights of the first layer, we see that their dynamics are simply given by a rescaled loss gradient. Now, let us look at the evolution of the

activations of the first layer in terms of their weights

$$\frac{d\mathbf{h}_\mu^{(1)}}{dt} = \frac{1}{N^{a_1}} \frac{d\mathbf{W}^{(1)}}{dt} \frac{1}{\sqrt{D}} \mathbf{x}_\mu \tag{80}$$

$$= \frac{\eta_0 \gamma_0}{N^{c+2a_1-d+1/2} s(\boldsymbol{\theta})} \frac{1}{DP} \sum_{\nu=1}^P \Delta_\nu \mathbf{g}_\nu^{(1)} \mathbf{x}_\nu \cdot \mathbf{x}_\mu \tag{81}$$

$$= \frac{\eta_0}{N^{c+2a_1-d+1/2} s(\boldsymbol{\theta})} \frac{1}{P} \sum_{\nu=1}^P \Delta_\nu \mathbf{g}_\nu^{(1)} \Phi_{\mu\nu}^{(0)}. \tag{82}$$

Given the results of Eqs. 41-42 & 47, we obtain $c + 2a_1 - d + 1/2 = \min(0, 2d)$. Given the Desideratum 2 constraint $c + 2a_1 = \min(0, 2d)$ (§A.4.3), we get

$$\boxed{d = 1/2}. \tag{83}$$

This has important implications for the previous desideratum (§A.4.3). Recall that Desideratum 2 gave constraints for two terms, a rescaled NTK term $q$ and a PC-specific gradient term $r$. If we compare the scaling of these two terms (Eqs. 66 & 73, respectively), it is clear that $d = 1/2$ makes $q$ the dominant term with respect to $N$, by a factor of $1/2$. This implies that, in order for the second desideratum to be satisfied, $q$ must be $\Theta_N(1)$, and $r$ must vanish at a rate of $\Theta(N^{-1/2})$.

Resuming our derivation and moving on to the next layer, we again start by writing the dynamics of the weights

$$\frac{d\mathbf{W}^{(\ell)}}{dt} = -\eta \frac{\partial \mathcal{F}_\nu^*}{\partial \mathbf{W}^{(\ell)}} = -\eta \frac{1}{s(\boldsymbol{\theta})P} \sum_{\nu=1}^P \frac{\partial \mathcal{L}_\nu}{\partial \mathbf{W}^{(\ell)}} + \eta \frac{1}{P} \sum_{\nu=1}^P \frac{\mathcal{L}_\nu}{s(\boldsymbol{\theta})^2} \frac{\partial s(\boldsymbol{\theta})}{\partial \mathbf{W}^{(\ell)}} \tag{84}$$

$$= \frac{\eta_0 \gamma^2}{N^c} \frac{1}{s(\boldsymbol{\theta})P} \sum_{\nu=1}^P \frac{\Delta_\nu}{\gamma} \frac{\partial h_\nu^{(L)}}{\partial \mathbf{h}_\nu^{(\ell)}} \frac{\partial \mathbf{h}_\nu^{(\ell)}}{\partial \mathbf{W}^{(\ell)}} + \frac{\eta_0 \gamma^2}{N^c} \frac{1}{P} \sum_{\nu=1}^P \frac{\mathcal{L}_\nu}{s(\boldsymbol{\theta})^2} \frac{\partial s(\boldsymbol{\theta})}{\partial \mathbf{W}^{(\ell)}} \tag{85}$$

$$= \frac{\eta_0 \gamma_0}{N^{c+a_\ell-d+1/2}} \frac{1}{s(\boldsymbol{\theta})P} \sum_{\nu=1}^P \Delta_\nu \mathbf{g}_\nu^{(\ell)} (\mathbf{h}_\nu^{(\ell-1)})^\top + \frac{\eta_0 \gamma^2}{N^c} \frac{1}{P} \sum_{\nu=1}^P \frac{\mathcal{L}_\nu}{s(\boldsymbol{\theta})^2} \frac{\partial s(\boldsymbol{\theta})}{\partial \mathbf{W}^{(\ell)}}. \tag{86}$$

Note that now, for $\ell > 1$, we have an extra energy gradient term. To simplify notation, let $\mathbf{D}^{(\ell)} := \frac{\eta_0 \gamma^2}{N^c} \frac{1}{P} \sum_\nu \frac{\mathcal{L}_\nu}{s(\boldsymbol{\theta})^2} \frac{\partial s}{\partial \mathbf{W}^{(\ell)}}$. The evolution of the activations is then

$$\frac{d\mathbf{h}_\mu^{(\ell)}}{dt} = \frac{1}{N^{a_\ell}} \frac{d\mathbf{W}^{(\ell)}}{dt} \mathbf{h}_\mu^{(\ell-1)} + \frac{1}{N^{a_\ell}} \mathbf{W}^{(\ell)} \frac{d\mathbf{h}_\mu^{(\ell-1)}}{dt} \tag{87}$$

$$= \frac{\eta_0 \gamma_0}{N^{c+2a_\ell-d+1/2}} \frac{1}{s(\boldsymbol{\theta})P} \sum_{\nu=1}^P \Delta_\nu \mathbf{g}_\nu^{(\ell)} \frac{N}{N} \mathbf{h}_\nu^{(\ell-1)} \cdot \mathbf{h}_\nu^{(\ell-1)} + \frac{1}{N^{a_\ell}} \mathbf{D}^{(\ell)} \mathbf{h}_\mu^{(\ell-1)} + \frac{1}{N^{a_\ell}} \mathbf{W}^{(\ell)} \frac{d\mathbf{h}_\mu^{(\ell-1)}}{dt} \tag{88}$$

$$= \frac{\eta_0 \gamma_0}{N^{c+2a_\ell-d-1/2}} \frac{1}{s(\boldsymbol{\theta})P} \sum_{\nu=1}^P \Delta_\nu \mathbf{g}_\nu^{(\ell)} \Phi_{\mu\nu}^{(\ell-1)} + \frac{1}{N^{a_\ell}} \mathbf{D}^{(\ell)} \mathbf{h}_\mu^{(\ell-1)} + \frac{1}{N^{a_\ell}} \mathbf{W}^{(\ell)} \frac{d\mathbf{h}_\mu^{(\ell-1)}}{dt}. \tag{89}$$

The first term implies that $c + 2a_\ell - d - 1/2 = \min(0, 2d)$ which, under the Desideratum 2 constraint (Eq. 66), again gives

$$\boxed{d = 1/2}. \tag{90}$$

Using Eqs. 55 & 47, and recalling that the loss and the activations of the previous layer are $\Theta_N(1)$ by the forward pass constraint (Desideratum 1), the entries of the $\mathbf{D}$ term scale as follows

$$\frac{1}{N^{a_\ell}} \sum_k^N D_{ik}^{(\ell)} h_{\mu,k}^{(\ell-1)} = \frac{\eta_0 \gamma^2}{N^{c+a_\ell}} \frac{\mathcal{L}_\nu}{s(\boldsymbol{\theta})^2} \sum_k^N \frac{\partial s(\boldsymbol{\theta})}{\partial W_{ik}^{(\ell)}} h_{\mu,k}^{(\ell-1)} \tag{91}$$

$$= \Theta \left( \frac{1}{N^{c+a_\ell-2d}(1 + N^{2d})^2} N^{1-2d-a_\ell-1/2} \right) \tag{92}$$

$$= \Theta \left( \frac{1}{N^{c+2a_\ell-1/2}(1 + N^{-2d})^2} \right), \tag{93}$$

giving the constraint

$$\boxed{c + 2a_\ell - 1/2 = \min(0, 2d)}. \tag{94}$$

Similar to the PC-specific term $r$ in the network function's evolution, the Desideratum 2 constraint (Eq. 66) implies that an entry of the $\mathbf{D}$ term will vanish as $\Theta(N^{-1/2})$. Finally, note that the covariance of the third term of Eq. 89 is $\Theta_N(1)$ by the forward pass desideratum, since we just established that $dh_{\mu,i}^{(\ell-1)}/dt = \Theta_N(1)$.

### A.4.5. COMBINING CONSTRAINTS

Putting together the constraints derived from all the desiderata (§A.4.1-A.4.4), we arrive at the following parameterisations:

1. Preactivations are $\Theta_N(1)$ at initialisation: $2a_\ell + b_\ell = 1$ for $\ell \in \{2, \dots, L\}$, and $2a_1 + b_1 = 0$.

2. Networks predictions are $\Theta_N(1)$ during training: $c + 2a_\ell = 1$ for $\ell \in \{2, \dots, L\}$, and $c + 2a_1 = 0$.

3. Features evolve in $\Theta_N(1)$ time during training: $d = 1/2$.

This is the same set of one-dimensional solutions derived for BP, thus concluding the proof of Theorem 1. See Figure A.5 for a visual illustration.

### A.5. Derivation of depth scalings for linear residual PCNs

Following the derivation of the width scalings in §A.4, this section derives the depth scaling exponent $\alpha$ that satisfies the 2 Desiderata described in §4, proving Theorem 2. Following previous work (Yang et al., 2023; Bordelon et al., 2023), we fix the width parameterisation to the mean-field (see Table 2) and extend it to residual networks as follows:[5]

$$f_\mu = \frac{1}{\gamma_0 \sqrt{N}} h_\mu^{(L)}, \tag{95}$$

$$h_\mu^{(L)} = \frac{1}{\sqrt{N}} \mathbf{w}^{(L)} \mathbf{h}_\mu^{(L-1)}, \tag{96}$$

$$\mathbf{h}_\mu^{(\ell)} = \left( \mathbf{I} + \frac{1}{L^\alpha \sqrt{N}} \mathbf{W}^{(\ell)} \right) \mathbf{h}_\mu^{(\ell-1)}, \tag{97}$$

$$\mathbf{h}_\mu^{(1)} = \frac{1}{\sqrt{D}} \mathbf{W}^{(1)} \mathbf{x}_\mu, \tag{98}$$

where $\alpha$ is some depth-dependent exponent scaling the residual branches. Recall that under our (mean-field) parameterisation, all the weights are initialised as $\theta_i \sim \mathcal{N}(0, 1)$, and $\gamma = \gamma_0 \sqrt{N}$ since $d = 1/2$. We will again consider gradient flow on the equilibrated energy (Eq. 5), with learning rate $\eta = \eta_0 \gamma^2 N^{-c} = \eta_0 \gamma_0^2 N$ since $c = 0$.

### A.5.1. DESIDERATUM 4: PREACTIVATIONS ARE $\Theta_L(1)$ AT INITIALISATION

Similar to the first desideratum (§A.4.1), we require that the mean-squared values of the residual stream variables (Eq. 18) are $\Theta_L(1)$ at initialisation. Similar derivations have been performed in previous work (Hayou & Yang, 2023; Hayou et al., 2021), but we again include example calculations for completeness.

We showed in §A.4.1 that the mean of the preactivations at each layer is zero (because of the zero-mean weight initialisation), and this fact does not change for residual networks. The covariance admits the following known recursion:

$$\left\langle h_{\mu,i}^{(\ell)} h_{\nu,j}^{(\ell)} \right\rangle = \left\langle h_{\mu,i}^{(\ell-1)} h_{\nu,j}^{(\ell-1)} \right\rangle + \frac{1}{L^{2\alpha} N} \sum_k^N \sum_{k'}^N \left\langle W_{ik}^{(\ell)}(0) W_{jk'}^{(\ell)}(0) \right\rangle \left\langle h_{\mu,k}^{(\ell-1)} h_{\nu,k'}^{(\ell-1)} \right\rangle \tag{99}$$

$$= \left\langle h_{\mu,i}^{(\ell-1)} h_{\nu,j}^{(\ell-1)} \right\rangle + \delta_{ij} \frac{1}{L^{2\alpha}} \left\langle h_{\mu,1}^{(\ell-1)} h_{\nu,1}^{(\ell-1)} \right\rangle \tag{100}$$

$$= \delta_{ij} \left\langle h_{\mu,i}^{(\ell-1)} h_{\nu,j}^{(\ell-1)} \right\rangle \left( 1 + \frac{1}{L^{2\alpha}} \right), \tag{101}$$

---

[5]Note that the specific choice of width-stable and feature-learning parameterisation (Eq. 15) does not affect the results of the depth scaling analysis, since they all satisfy the same desiderata.

where, as in §A.4.1, we use the fact that neurons are i.i.d. are initialisation. Unrolling this recursion up to the last residual activation $h_{\mu,i}^{(L-1)}$, we get

$$\left\langle h_{\mu,i}^{(L-1)} h_{\nu,j}^{(L-1)} \right\rangle = \delta_{ij} \Phi_{\mu\nu}^{(0)} \left( 1 + \frac{1}{L^{2\alpha}} \right)^{L-2}, \tag{102}$$

where recall from Eq. 38 that $\Phi_{\mu\nu}^{(0)}$ is the normalised feature kernel of the data. To keep this $\Theta_L(1)$, we need

$$\boxed{\alpha \geq 1/2}. \tag{103}$$

Note that for $\alpha = 1/2$ each residual activation has exactly a $\Theta(1/L)$ contribution, while for $\alpha > 1/2$ their contribution vanishes, and the covariance reduces to the data kernel. Also note that, for MLPs, $\alpha = 0$ is the only (critical) value for which the forward pass is stable (Poole et al., 2016; Schoenholz et al., 2016).

### A.5.2. USEFUL ASYMPTOTIC RESULTS FOR RESIDUAL NETWORKS

Before moving on to the next desideratum, it is useful to extend some of the asymptotic results derived for MLPs in §A.4.2 to residual networks. In particular, we will extend the PC-specific results about the equilibrated energy rescaling and its parameter derivative.

**Equilibrated energy rescaling.** First, under our considered residual network parameterisation (Eqs. 95-98), the equilibrated energy rescaling takes the following form (Innocenti et al., 2025):

$$s(\boldsymbol{\theta}) = 1 + \frac{1}{\gamma_0^2 N^2} \left( \|\mathbf{w}^{(L)}\|^2 + \sum_{\ell=2}^{L-1} \left\| \mathbf{p}^{(L:\ell)} \right\|^2 \right), \tag{104}$$

$$\mathbf{p}^{(L:\ell)} := \mathbf{w}^{(L)} \prod_{\ell}^{L-1} \left( \mathbf{I}_N + \frac{1}{L^\alpha \sqrt{N}} \mathbf{W}^{(\ell)} \right), \tag{105}$$

where $\mathbf{p}^{(L:\ell)}$ denotes the $\ell$th residual path from $\ell > 1$ to $L$, and we define the matrix product as $\prod_{\ell=1}^{L} \mathbf{A}^{(\ell)} = \mathbf{A}^{(L)} \mathbf{A}^{(L-1)} \ldots \mathbf{A}^{(1)}$. The squared norm of the output weights scales as $\langle \|\mathbf{w}^{(L)}\|^2 \rangle = \Theta(N)$, and from the previous forward pass constraint $\alpha \geq 1/2$ (§A.5.1), the sum of the squared residual norms is $\langle \sum_{\ell=2}^{L-1} \|\mathbf{p}^{(L:\ell)}\|^2 \rangle = \sum_{\ell=2}^{L-1} \Theta(N) = \Theta(LN)$. We therefore get

$$s(\boldsymbol{\theta}) = 1 + \frac{1}{\gamma_0^2 N^2} \left( \Theta(N) + \Theta(LN) \right) \tag{106}$$

$$= 1 + \Theta \left( \frac{1}{N} + \frac{L}{N} \right) = 1 + \Theta \left( L/N \right), \tag{107}$$

which we see asymptotically scales in the same way as for MLPs (Eq. 47).

**Rescaling derivative with respect to hidden layer weights, $\partial s / \partial \mathbf{W}^{(\ell)}$.** We will also need the derivative of the rescaling with respect to any hidden weight matrix (cf. Eq. 48). We first define

$$\mathbf{P}^{(\ell:k)} := \prod_{k}^{\ell} \left( \mathbf{I} + \frac{1}{L^\alpha \sqrt{N}} \mathbf{W}^{(k)} \right), \tag{108}$$

with boundary conditions $\mathbf{P}^{(L-1:L)} = \mathbf{I}$ and $\mathbf{P}^{(\ell-1:\ell)} = \mathbf{I}$. The path from the output to the $\ell$th residual layer can now also be expressed as $\mathbf{p}^{(L:\ell)} = \mathbf{w}^{(L)} \mathbf{P}^{(L-1:\ell)}$. Given these, the derivative of the rescaling with respect to any hidden weight matrix for $\ell \in \{2, \ldots, L-1\}$ (cf. Eq. 48) is

$$\frac{\partial s}{\partial \mathbf{W}^{(\ell)}} = \frac{2}{\gamma_0^2 N^2 L^\alpha \sqrt{N}} \sum_{k=2}^{\ell} \left( \mathbf{w}^{(L)} \mathbf{P}^{(L-1:\ell+1)} \right)^\top \left( \mathbf{w}^{(L)} \mathbf{P}^{(L-1:k)} \right) \left( \mathbf{P}^{(\ell-1:k)} \right)^\top, \tag{109}$$

where as in the MLP case $\left(\mathbf{w}^{(L)}\mathbf{P}^{(L-1:\ell+1)}\right)^{\top} \in \mathbb{R}^{N \times 1}$, $\mathbf{w}^{(L)}\mathbf{P}^{(L-1:k)} \in \mathbb{R}^{1 \times N}$, and $\left(\mathbf{P}^{(\ell-1:k)}\right)^{\top} \in \mathbb{R}^{N \times N}$. The matrix product has the same structure as before (see Eq. 53), namely $\mathbf{M}^{(\ell,k)} = \mathbf{a}\mathbf{b}^{T}\mathbf{C}$, with $\mathbf{a} = \left(\mathbf{w}^{(L)}\mathbf{P}^{(L-1:\ell+1)}\right)^{\top}$, $\mathbf{b}^{\top} = \mathbf{w}^{(L)}\mathbf{P}^{(L-1:k)}$ and $\mathbf{C} = \left(\mathbf{P}^{(\ell-1:k)}\right)^{\top}$. In addition, under our parameterisation, all the three terms of the matrix product are $\Theta_N(1)$, so $M_{ij} = \Theta_N(1)$. Therefore,

$$\frac{\partial s}{\partial W_{ij}^{(\ell)}} = \frac{2}{\gamma_0^2 N^2 L^{\alpha}\sqrt{N}} \sum_{k=2}^{\ell} \Theta_N(1) = \Theta(L^{1-\alpha}N^{-5/2}), \quad N, L \to \infty. \tag{110}$$

### A.5.3. DESIDERATUM 5: FEATURES EVOLVE IN $\Theta_L(1)$ TIME DURING TRAINING

Similar to the third desideratum (§A.4.4), we also require that $dh_i^{(\ell)}/dt = \Theta_L(1)$. Recall that under our parameterisation $\gamma = \gamma_0\sqrt{N}$ and $\eta = \eta_0\gamma_0^2 N$. As before, it is useful to first write the dynamics of the weights

$$\frac{d\mathbf{W}^{(\ell)}}{dt} = -\eta\frac{\partial\mathcal{F}_{\nu}^{*}}{\partial\mathbf{W}^{(\ell)}} = \frac{\eta_0\gamma_0^2 N}{s(\boldsymbol{\theta})}\frac{1}{P}\sum_{\nu=1}^{P}\frac{\Delta_{\nu}}{\gamma_0\sqrt{N}}\frac{\partial h_{\nu}^{(L)}}{\partial\mathbf{h}_{\nu}^{(\ell)}}\frac{\partial\mathbf{h}_{\nu}^{(\ell)}}{\partial\mathbf{W}^{(\ell)}} + \frac{\eta_0\gamma_0^2 N}{s(\boldsymbol{\theta})^2}\frac{1}{P}\sum_{\nu=1}^{P}\mathcal{L}_{\nu}\frac{\partial s(\boldsymbol{\theta})}{\partial\mathbf{W}^{(\ell)}} \tag{111}$$

$$= \frac{\eta_0\gamma_0}{N^{-1/2}s(\boldsymbol{\theta})}\frac{1}{P}\sum_{\nu=1}^{P}\Delta_{\nu}\frac{\sqrt{N}}{\sqrt{N}}\frac{\partial h_{\nu}^{(L)}}{\partial\mathbf{h}_{\nu}^{(\ell)}}\frac{1}{L^{\alpha}\sqrt{N}}(\mathbf{h}_{\nu}^{(\ell-1)})^{\top} + \frac{\eta_0\gamma_0^2 N}{s(\boldsymbol{\theta})^2}\frac{1}{P}\sum_{\nu=1}^{P}\mathcal{L}_{\nu}\frac{\partial s(\boldsymbol{\theta})}{\partial\mathbf{W}^{(\ell)}} \tag{112}$$

$$= \frac{\eta_0\gamma_0}{\sqrt{N}s(\boldsymbol{\theta})}\frac{1}{P}\sum_{\nu=1}^{P}\Delta_{\nu}\mathbf{g}_{\nu}^{(\ell)}\frac{1}{L^{\alpha}}(\mathbf{h}_{\nu}^{(\ell-1)})^{\top} + \frac{\eta_0\gamma_0^2 N}{s(\boldsymbol{\theta})^2}\frac{1}{P}\sum_{\nu=1}^{P}\mathcal{L}_{\nu}\frac{\partial s(\boldsymbol{\theta})}{\partial\mathbf{W}^{(\ell)}}. \tag{113}$$

As in §A.4.4, let $\mathbf{D}^{(\ell)} := \frac{\eta_0\gamma_0^2 N}{s(\boldsymbol{\theta})^2}\frac{1}{P}\sum_{\nu}^{P}\mathcal{L}_{\nu}\frac{\partial s}{\partial\mathbf{W}^{(\ell)}}$ to keep the notation compact. The evolution of any residual activation for $\ell \in \{2, \ldots, L-1\}$ is then

$$\frac{d\mathbf{h}_{\mu}^{(\ell)}}{dt} = \frac{1}{L^{\alpha}\sqrt{N}}\frac{d\mathbf{W}^{(\ell)}}{dt}\mathbf{h}_{\mu}^{(\ell-1)} + \left(\mathbf{I} + \frac{1}{L^{\alpha}\sqrt{N}}\mathbf{W}^{(\ell)}\right)\frac{d\mathbf{h}_{\mu}^{(\ell-1)}}{dt} \tag{114}$$

$$= \frac{d\mathbf{h}_{\mu}^{(\ell-1)}}{dt} + \frac{1}{L^{\alpha}\sqrt{N}}\left(\frac{d\mathbf{W}^{(\ell)}}{dt}\mathbf{h}_{\mu}^{(\ell-1)} + \mathbf{W}^{(\ell)}\frac{d\mathbf{h}_{\mu}^{(\ell-1)}}{dt}\right) \tag{115}$$

$$= \frac{d\mathbf{h}_{\mu}^{(\ell-1)}}{dt} + \frac{1}{L^{\alpha}\sqrt{N}}\left[\left(\frac{\eta_0\gamma_0}{\sqrt{N}s(\boldsymbol{\theta})}\frac{1}{P}\sum_{\nu=1}^{P}\Delta_{\nu}\mathbf{g}_{\nu}^{(\ell)}\frac{1}{L^{\alpha}}(\mathbf{h}_{\nu}^{(\ell-1)})^{\top} + \mathbf{D}^{(\ell)}\right)\mathbf{h}_{\mu}^{(\ell-1)} + \mathbf{W}^{(\ell)}\frac{d\mathbf{h}_{\mu}^{(\ell-1)}}{dt}\right] \tag{116}$$

$$= \frac{d\mathbf{h}_{\mu}^{(\ell-1)}}{dt} + \frac{1}{L^{2\alpha}}\frac{\eta_0\gamma_0}{s(\boldsymbol{\theta})}\frac{1}{P}\sum_{\nu=1}^{P}\Delta_{\nu}\mathbf{g}_{\nu}^{(\ell)}\Phi_{\mu\nu}^{(\ell-1)} + \frac{1}{L^{\alpha}\sqrt{N}}\left(\mathbf{D}^{(\ell)}\mathbf{h}_{\mu}^{(\ell-1)} + \mathbf{W}^{(\ell)}\frac{d\mathbf{h}_{\mu}^{(\ell-1)}}{dt}\right) \tag{117}$$

$$= \frac{d\mathbf{h}_{\mu}^{(\ell-1)}}{dt} + \frac{1}{L^{2\alpha}}\frac{\eta_0\gamma_0}{s(\boldsymbol{\theta})}\frac{1}{P}\sum_{\nu=1}^{P}\Delta_{\nu}\mathbf{g}_{\nu}^{(\ell)}\Phi_{\mu\nu}^{(\ell-1)} \tag{118}$$

$$+ \frac{\eta_0\gamma_0^2}{L^{\alpha}N^{-1/2}s(\boldsymbol{\theta})^2}\frac{1}{P}\sum_{\nu=1}^{P}\mathcal{L}_{\nu}\frac{\partial s}{\partial\mathbf{W}^{(\ell)}}\mathbf{h}_{\mu}^{(\ell-1)} + \frac{1}{L^{\alpha}\sqrt{N}}\mathbf{W}^{(\ell)}\frac{d\mathbf{h}_{\mu}^{(\ell-1)}}{dt}. \tag{119}$$

Unrolling this recursion as in §A.5.1, and ignoring the last, higher-order term, we get

$$\frac{d\mathbf{h}_{\mu}^{(L-1)}}{dt} \approx \frac{d\mathbf{h}_{\mu}^{(1)}}{dt} + \sum_{\ell=2}^{L-1}\frac{1}{L^{2\alpha}}\left(\frac{\eta_0\gamma_0}{s(\boldsymbol{\theta})}\frac{1}{P}\sum_{\nu=1}^{P}\Delta_{\nu}\mathbf{g}_{\nu}^{(\ell)}\Phi_{\mu\nu}^{(\ell-1)}\right) + \sum_{\ell=2}^{L-1}\frac{1}{L^{\alpha}\sqrt{N}}\mathbf{D}^{(\ell)}\mathbf{h}_{\mu}^{(\ell-1)} \tag{120}$$

$$\approx \frac{d\mathbf{h}_{\mu}^{(1)}}{dt} + \sum_{\ell=2}^{L-1}\frac{1}{L^{2\alpha}}\left(\frac{\eta_0\gamma_0}{s(\boldsymbol{\theta})}\frac{1}{P}\sum_{\nu=1}^{P}\Delta_{\nu}\mathbf{g}_{\nu}^{(\ell)}\Phi_{\mu\nu}^{(\ell-1)}\right) + \frac{\eta_0\gamma_0^2}{P}\sum_{\nu=1}^{P}\mathcal{L}_{\nu}\left(\frac{1}{L^{\alpha}N^{-1/2}s(\boldsymbol{\theta})^2}\sum_{\ell=2}^{L-1}\frac{\partial s}{\partial\mathbf{W}^{(\ell)}}\right)\mathbf{h}_{\mu}^{(\ell-1)}. \tag{121}$$

Using previous results including Eq. 110, this scales as

$$\frac{d\mathbf{h}_\mu^{(L-1)}}{dt} = \Theta\left(\frac{L^{1-2\alpha}}{(1+N^{-1})}\right) + \Theta\left(\frac{L^{1-\alpha}}{N^{-1/2}(1+N^{-1})^2}\frac{L^{1-\alpha}}{N^{5/2}}\right) \tag{122}$$

$$= \Theta\left(L^{1-2\alpha}\right) + \Theta\left(L^{2-2\alpha}N^{-2}\right). \tag{123}$$

As shown in previous work (Yang et al., 2023; Bordelon et al., 2023), we therefore find that

$$\boxed{\alpha = 1/2} \tag{124}$$

in order for the residual feature updates to be $\Theta_L(1)$. It follows that the second, PC-specific term scales as $\Theta(L/N^2)$. This completes our proof of Theorem 2.

### A.6. Parameterisation extensions

Because we showed that the set of stable and feature-learning parameterisations for linear PCNs is the same as for BP (Theorems 1-2), it is relatively straightforward to extend our results to closely related architectures such as CNNs and other optimisers such as Adam (Kingma & Ba, 2014). For Adam, since it is approximately scale-invariant to the gradient, we would also need to further scale the learning rate (of hidden matrices) $\eta$ by $N^{-1/2}L^{\alpha-1}$ to keep the feature updates stable with the width and depth (Yang & Littwin, 2023; Bordelon et al., 2024; Dey et al., 2025).

#### A.6.1. CNNs

For CNNs, we can adjust the scalings (Table 1) by essentially taking the number of channels as the width. The mean-field-parameterised PC energy for the CNN architecture presented in Figure 4 is as follows:

$$\mathcal{F}(\mathbf{Z},\boldsymbol{\theta}) = \frac{1}{2P}\sum_{\mu=1}^{P}\Big($$

$$\|\mathbf{Z}_\mu^{(1)} - \underbrace{\frac{1}{\sqrt{D}}\mathbf{W}^{(1)}\mathbf{X}_\mu}_{\text{2D Conv.}}\|_F^2 + \|\mathbf{Z}_\mu^{(2)} - \underbrace{(\frac{1}{\sqrt{NL}}\mathbf{W}^{(2)}\phi(\mathbf{Z}_\mu^{(1)}) + \mathbf{Z}_\mu^{(1)})}_{\text{Res. block}}\|_F^2 + \|\mathbf{Z}_\mu^{(3)} - \underbrace{\mathbf{W}^{(3)}\mathbf{Z}_\mu^{(2)}}_{\text{Avg. pool.}}\|_F^2 \quad \text{[Stage 1]}$$

$$+ \|\mathbf{Z}_\mu^{(4)} - \frac{1}{\sqrt{N}}\mathbf{W}^{(4)}\mathbf{Z}_\mu^{(3)}\|_F^2 + \|\mathbf{Z}_\mu^{(5)} - (\frac{1}{\sqrt{NL}}\mathbf{W}^{(5)}\phi(\mathbf{Z}_\mu^{(4)}) + \mathbf{Z}_\mu^{(4)})\|_F^2 + \|\mathbf{Z}_\mu^{(6)} - \mathbf{W}^{(6)}\mathbf{Z}_\mu^{(5)}\|_F^2 \quad \text{[Stage 2]}$$

$$+ \|\mathbf{Z}_\mu^{(7)} - \frac{1}{\sqrt{N}}\mathbf{W}^{(7)}\mathbf{Z}_\mu^{(6)}\|_F^2 + \|\mathbf{Z}_\mu^{(8)} - (\frac{1}{\sqrt{NL}}\mathbf{W}^{(8)}\phi(\mathbf{Z}_\mu^{(7)}) + \mathbf{Z}_\mu^{(7)})\|_F^2 + \|\mathbf{Z}_\mu^{(9)} - \mathbf{W}^{(9)}\mathbf{Z}_\mu^{(8)}\|_F^2 \quad \text{[Stage 3]}$$

$$+ \mathcal{L}_{\text{CE}}\big(\underbrace{\frac{1}{C_{\text{out}}H'W'}\mathbf{W}^{(10)}\text{vec}(\mathbf{Z}_\mu^{(9)})}_{\text{Readout}}, \mathbf{y}_\mu\big)$$

$$\Big). \tag{125}$$

This architecture can be described as a "3-stage ResNet", where each stage applies:

1. a 2D convolution $\mathbf{W}^{(\ell)} \in \mathbb{R}^{C_{\text{out}}\times(k^2 C_{\text{in}})}$ with kernel size $k$ and input and output channels $(C_{\text{in}}, C_{\text{out}})$;

2. one residual block consisting of a nonlinearity $\phi(\cdot)$ plus convolution; and

3. an average pooling operation expressed as a fixed sparse linear transformation $\mathbf{W}^{(\ell)}$ for $\ell \in \{3, 6, 9\}$, with kernel size $k = 2$ and stride $s = 2$,

before a final fully connected layer $\mathbf{W}^{(10)} \in \mathbb{R}^{C\times(C_{\text{out}}H'W')}$, where $C$ is the number of classes, and $H'$ and $W'$ are the input's final height and width dimensions, respectively. For the experiment in Figure 4, we used Tanh as nonlinearity. As before, all the weights are initialised from a standard Gaussian, $W_{ij}^{(\ell)} \sim \mathcal{N}(0,1)$. The last term of Eq. 125 is the cross-entropy loss, i.e. $\mathcal{L}_{\text{CE}}(\hat{\mathbf{y}}, \mathbf{y}) = -\sum_{c=1}^{C} y_c[\log\text{softmax}(\hat{\mathbf{y}})]_c$. All convolutions have unit stride and padding. The

input image $\mathbf{X}_\mu$ is transformed to have size $(k^2 C_{\text{in}}) \times HW$, so that 2D convolutions can be expressed as simple matrix multiplications. Note that here $D = k^2 C_{\text{in}}$. As in the experiment in Figure 4, both the kernel size and the number of output channels are kept constant across all convolutions, giving an effective model width of $N = k^2 C_{\text{out}}$.[6] The depth scaling factor $L$ is given by the total number of residual blocks (which is what is theoretically taken to infinity), in this case 3 (with one block in each stage). The experiments in Figure 4 add non-residual layers to the depth scaling, following previous work (Bordelon et al., 2023).

### A.6.2. TRANSFORMERS

Given an input sequence $\mathbf{X}_\mu \in \mathbb{R}^{T \times V}$, with $T$ as its length and $V$ as the vocabulary size, the PC energy of the transformer tested in Figure 4 is given by:

$$
\begin{aligned}
\mathcal{F}(\mathbf{Z}, \boldsymbol{\theta}) = \frac{1}{2TP} \sum_{\mu=1}^{P} \Big( & \\
& \|\mathbf{Z}_\mu^{(1)} - (\mathbf{X}_\mu \mathbf{W}^{(\text{tok.})} + \mathbf{I}_T \mathbf{W}^{(\text{pos.})})\|_F^2 \qquad\qquad\qquad \text{[Embeddings]} \\
& + \sum_{\ell=2}^{L+\ell-1} \left\| \mathbf{Z}_\mu^{(\ell)} - \mathbf{Z}_\mu^{(\ell-1)} - \underbrace{\frac{1}{L}\text{MHA}(\mathbf{Z}_\mu^{(\ell-1)})}_{\text{Multi-head attention}} - \underbrace{\frac{1}{L}\text{MLP}\Big(\mathbf{Z}_\mu^{(\ell-1)} + \frac{1}{L}\text{MHA}(\mathbf{Z}_\mu^{(\ell-1)})\Big)}_{\text{Feedforward network}} \right\|_F^2 \quad \text{[Blocks]} \\
& + \mathcal{L}_{\text{CE}}\big(\frac{1}{N}\mathbf{W}^{(\text{out})}\mathbf{Z}_\mu^{(\text{out}-1)}\big), \mathbf{y}_\mu\big) \qquad\qquad\qquad\qquad \text{[LM head]} \\
\Big), &
\end{aligned}
\tag{126}
$$

where $\mathbf{W}^{(\text{tok.})} \in \mathbb{R}^{V \times N}$ and $\mathbf{W}^{(\text{pos.})} \in \mathbb{R}^{T \times N}$ are token and positional embedding matrices, respectively, with $N$ as the model width (also known as the embedding dimension) and $L$ as the number of transformer blocks. The multi-head attention (MHA) (Vaswani et al., 2017) subblock is given by

$$
\text{MHA}(\mathbf{Z}) = \text{concat}(\text{head}_1, \ldots, \text{head}_h) \frac{1}{\sqrt{N}} \mathbf{W}^{(O)}, \tag{127}
$$

$$
\text{head}_i(\mathbf{Z}) = \text{softmax}\left( \frac{(\frac{1}{\sqrt{d_i}}\mathbf{Z}\mathbf{W}^{(Q,i)})(\frac{1}{\sqrt{d_i}}\mathbf{Z}\mathbf{W}^{(K,i)})^\top}{d_i} \right) (\frac{1}{\sqrt{d_i}}\mathbf{Z}\mathbf{W}^{(V,i)}), \tag{128}
$$

where each head $i \in \{1, \ldots, h\}$ has query, key and value matrices $\mathbf{W}^{(Q,i)}, \mathbf{W}^{(K,i)}, \mathbf{W}^{(V,i)} \in \mathbb{R}^{T \times d_i}$, with head dimension $d_i = N/h$. The output projection matrix $\mathbf{W}^{(O)} \in \mathbb{R}^{T \times N}$ acts on the concatenated output of each head. The MLP subblock is then

$$
\text{MLP}(\mathbf{Z}) = \frac{1}{\sqrt{N}} \mathbf{W}^{(2)} \phi(\frac{1}{\sqrt{4N}} \mathbf{W}^{(1)} \mathbf{Z}), \tag{129}
$$

where following standard hyperparameters we assume that the first layer expands the model width by a factor of 4. For the experiment in Figure 4, we used GeLU as nonlinearity. All the weights are again initialised from a standard Gaussian. The overall parameterisation is adapted from Dey et al. (2025), and we use exactly the same learning rate scalings for Adam.

## A.7. Additional experiments

### A.7.1. NONLINEAR NETWORKS

Here we report some experiments with nonlinear networks trained on toy tasks, supporting our theory of linear networks.

---

[6]Note that, even though it is the number of channels that are increased with the depth in practice (and therefore assumed to grow to infinity), the kernel size ($k^2$) can have an impact in practice, and this is the reason why it is included in the scalings.

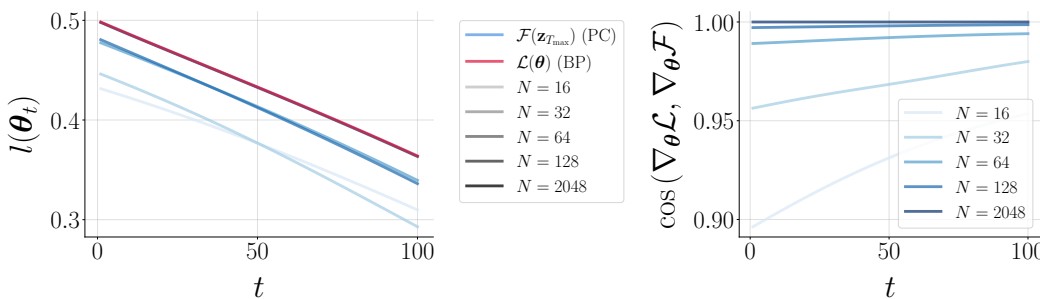

*Figure A.1.* **Under stable and feature-learning parameterisations, PC converges to BP for model width much larger than depth even on *nonlinear* networks.** With a similar setup to Figure 2, we plot results for nonlinear residual networks ($L = 5$) with Tanh as activation function. As in the linear case, we see that the PC energy at numerical convergence of the activities $\mathcal{F}(\mathbf{z}_{T_{max}})$ converges to the BP MSE loss (Eq. 1) for sufficiently large width (*Left*), resulting in PC computing the same gradients as BP (*Right*). See the next Figure for results with ReLU and Figures A.21-A.22 for similar results with MLPs.

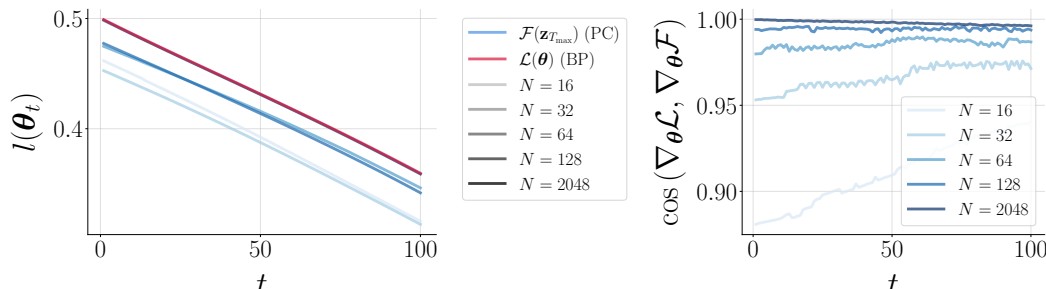

*Figure A.2.* **Results of Figure A.1 for ReLU.**

### A.7.2. LEARNING REGIMES

Here we show results related to §3.2. For both linear MLPs and residual networks, Figures A.3 & A.23 verify that, under width-stable feature-learning parameterisations, PC converges to BP at large width, recovering its learning regimes. On the other hand, Figures A.4 & A.24 show that, in the regime of larger depth than width, the fast PC saddle-to-saddle regime studied in Innocenti et al. (2024a) exists for MLPs but not residual networks, respectively.

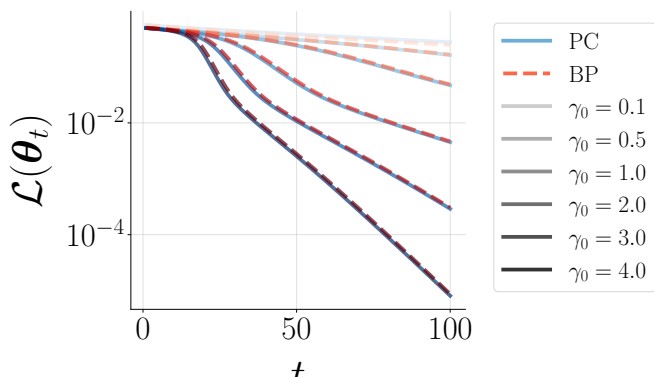

*Figure A.3.* **Stable feature-learning parameterisations of wide linear PCNs exhibit the same learning regimes (lazy vs rich) as BP at large width.** For the same setup of Figure 2, we fix the width at $N = 2048$ and compare the MSE loss (Eq. 1) of networks trained with both BP and PC for different values of the feature learning parameter $\gamma_0$ (Eq. 11). See Figure A.23 for similar results with residual networks.

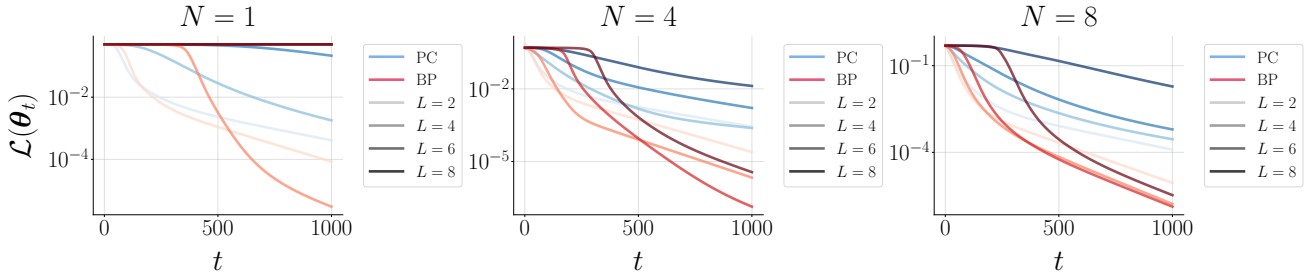

*Figure A.4.* **Under the SP, linear MLPs trained with PC have faster saddle escape dynamics for deeper than wide networks,** $L > N$. For the same setup of Figure 2, we plot the MSE loss (Eq. 1) for linear MLPs trained with BP and PC, varying both the width $N$ and depth $L$. We see that PC shows faster escape times of the origin saddle for $L > N$. As discussed in §3.2, this regime does not exist for linear residual networks, as shown in Figure A.24.

## A.8. Experimental details

Code to reproduce all the results is available at https://github.com/thebuckleylab/jpc/tree/main/experiments/limits_paper as part of a JAX library for training PCNs (Innocenti et al., 2024b). We always used no biases. For experiments with the SP, all networks used Kaiming Uniform $W_{ij}^{(\ell)} \sim \mathcal{U}(-1/\sqrt{d_{\text{in}}}, 1/\sqrt{d_{\text{in}}})$ as standard initialisation, with $d_{\text{in}}$ as the input dimension. Wherever we do not report error bars, results were consistent across different random seeds.

**Toy tasks.** We trained linear networks with BP and PC to predict binary labels $y \in \{-1, 1\}$ with 40-dimensional inputs $\mathbf{x} \in \mathbb{R}^{40}$ drawn from $\mathcal{N}(0, 1)$. Unless otherwise stated, all experiments on this task used full-batch GD with $P = 20$ samples, learning rate constant $\eta_0 = 0.025$ (Eq. 14), and feature learning parameter $\gamma_0 = 1$ (Eq. 11).

For Figure 2, we trained MLPs of depth $L = 5$ and varying widths $N \in \{2^i\}_{i=3}^{11}$. The plots show selected widths to aid visualisation. For both PC and BP, we ran experiments with the mean-field parameterisation (Figures 2 & A.18) and the SP (Figures A.19-A.20), as defined in Table 2. Note that for the mean-field parameterisation the output scaling is $\gamma = \gamma_0 \sqrt{N}$, and the learning rate is $\eta = \eta_0 \gamma^2 N^{-c} = \eta_0 \gamma_0^2 N$. For the experiments with residual networks (Figure A.17), we used $\alpha = 1/2$.

The theoretical infinite-width prediction and loss in Figure 2 were computed using the dynamical mean field theory developed by Bordelon & Pehlevan (2022b) (see in particular Appendix F). Under some approximations, this theory essentially allows one to integrate out the randomness over the weight initialisation and obtain a much lower-dimensional description of the network dynamics. For PC, we directly trained on (i.e. computed gradients of) the theoretical equilibrated energy (Eq. 5). Results with standard numerical optimisation of the activities (Eq. 3) are presented in §5 and detailed below.

For Figures A.3 & A.23, we fixed the width at $N = 2048$ and varied the feature learning parameter $\gamma_0 \in \{0.1, 0.5, 1, 2, 3, 4\}$ (Eq. 11). For Figures A.4 & A.24, we used the SP as in Innocenti et al. (2024a), and trained models of varying width and depth, with and without skips (respectively), for 1000 training iterations.

**Image classification tasks.** We trained networks to classify Fashion-MNIST and CIFAR-10. For the cosine similarity heatmaps (e.g. Figure 1) and rescaling plots (e.g. Figure 3), we trained mean-field-parameterised (see Table 2) linear and nonlinear networks, with and without skips, varying the width and depth. Cosine similarities were averaged over 3 random seeds. All experiments used Adam with batch size 64, and learning rate constant $\eta_0 = 1e^{-3}$.

**Nonlinear models.** For the results of Figure 4, we trained nonlinear residual networks, CNNs, and transformers with BP and PC. All experiments used 100 training steps, Adam and learning rate constant $\eta_0 = 1e^{-3}$. We plot cosine similarities during training for a representative run. The residual networks (MLPs) had 32 layers and Tanh as nonlinearity. For PC inference (Eq. 3), we used 100k iterations with step size $\beta = 0.3$ to ensure numerical convergence. All other training details for the residual MLPs, including the parameterisation (mean-field), were the same as in Figure 1.

The CNNs were trained on ImageNet with the cross-entropy loss and batch size 128. As explained in Figure 4, the width for these models is defined as the number of output channels times the kernel size, i.e. $N = k^2 C_{\text{out}}$. Following previous work (Bordelon et al., 2023), we kept the kernel size constant ($k = 3$) and varied the number of output channels $C_{\text{out}} \in \{2, 8, 16, 32\}$ of each convolutional layer. All the details of the architecture and parameterisation are given §A.6.1.

For PC inference, we used 200 iterations with step size $\beta = 0.3$.

The transformers had a nano-GPT-style architecture with 12 blocks, 8 attention heads and varying width or embedding dimension $N$, and were trained on the character-level Tiny Shakespeare dataset with the cross-entropy loss. We used batch size 64 and a sequence length of 32. Details of the architecture and parameterisation are given §A.6.2. The only key difference with the standard nano-GPT model was the removal of layer normalisation (Ba et al., 2016), which we found to break the convergence of PC to BP even at large width, perhaps because of its interference with the PC inference dynamics. For visualisation purposes, we report results without weight decay, since we found that convergence to BP was even faster with the width when using weight decay. For PC inference, we used 800 iterations with step size $\beta = 0.45$.

### A.9. Compute resources

The experiments with the widest and deepest models used a single NVIDIA RTX A6000 (48 GB VRAM). All other experiments were run on a CPU and took no more than a few hours, depending on the specific experiment.

### A.10. Supplementary figures

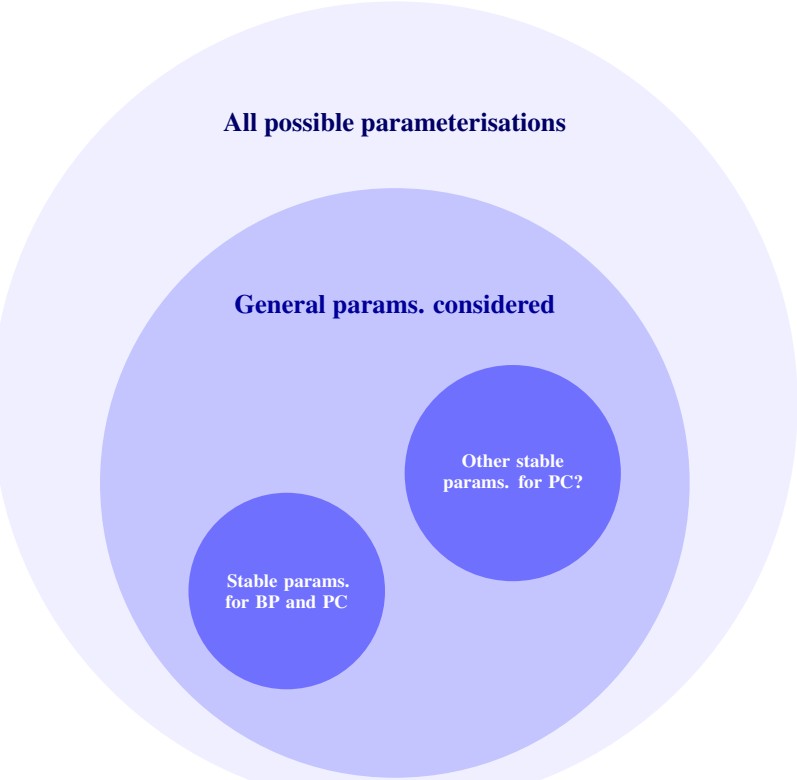

*Figure A.5.* **Venn diagram illustrating Theorems 1-2.** Following previous analyses for BP (Yang & Hu, 2021; Bordelon & Pehlevan, 2022b), we consider a specific set of general ("abcd") parameterisations of linear MLPs (Eqs. 7-10) and residual networks (Eq. 18), and show that the subset of stable (and feature-learning) parameterisations with respect to both model width and depth for PC is the same as for BP (Theorems 1-2). Within this subset, PC converges to BP for sufficiently large width (Corollaries 3.2-4.2). As discussed in §6, our results do not rule out other notions of stable and rich parameterisations (requiring different constraints), where PC might not converge to BP in some limit. Note also that, along with previous work, we do not consider all possible parameterisations.

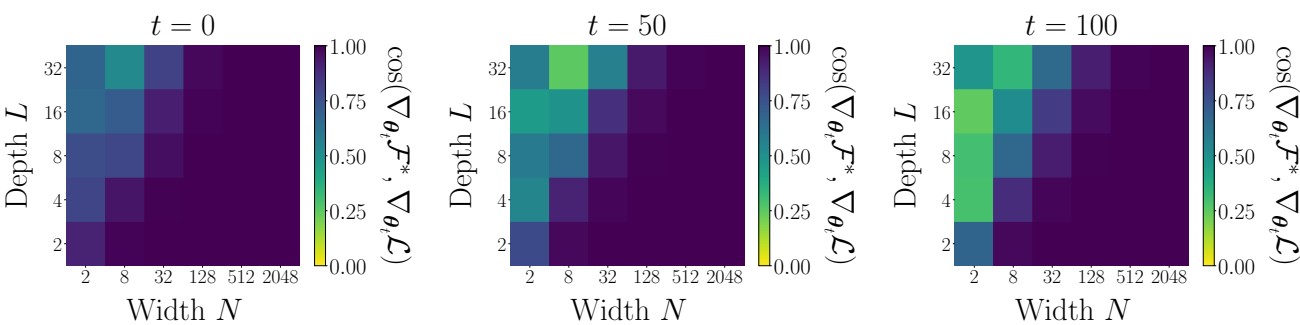

*Figure A.6.* **Results of Figure 1 for Fashion-MNIST.** A depth slice ($L = 32$) of these results is plotted in Figures A.11-A.12.

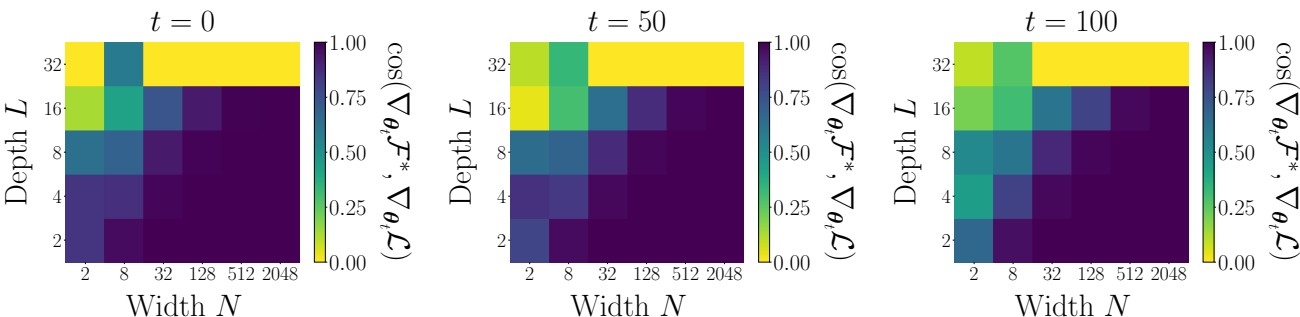

*Figure A.7.* **Results of Figure 1 for MLPs.** Unlike residual networks, the alignment between the PC and BP gradients breaks down at sufficiently large depth ($L = 32$) independent of the width, due the "fragility" of the MLP forward pass with depth as discussed in §3.2.

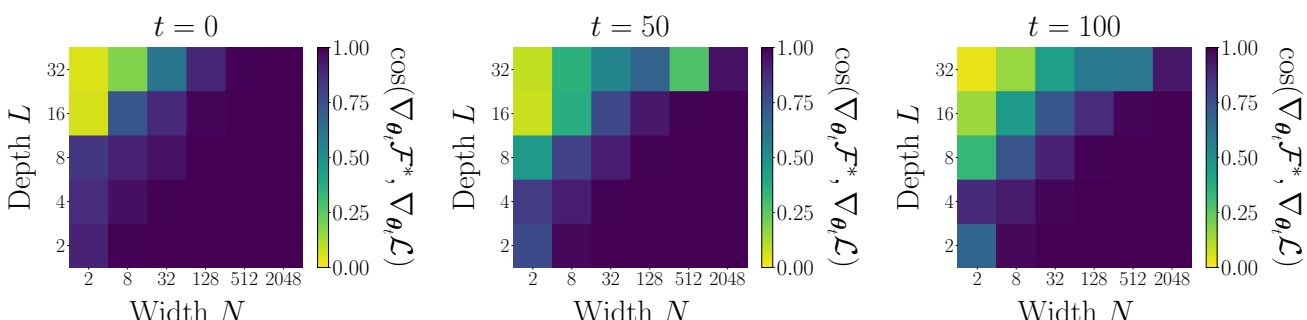

*Figure A.8.* **Results of the previous figure (A.7) for Fashion-MNIST.** Similar to the previous figure, we see that the PC gradients diverge from the BP gradients at large enough depth ($L = 32$), regardless of the width. Results for varying width at depth $L = 32$ are plotted in Figures A.13-A.14.

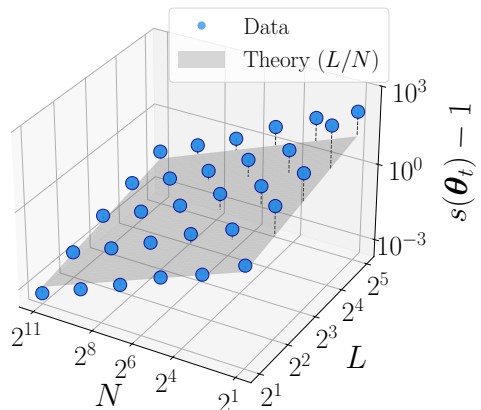

*Figure A.9.* **Results of Figure 3 for Fashion-MNIST.**

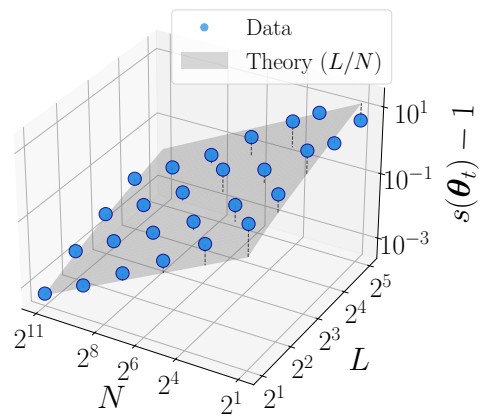

*Figure A.10.* **Results of Figure 3 for MLPs.**

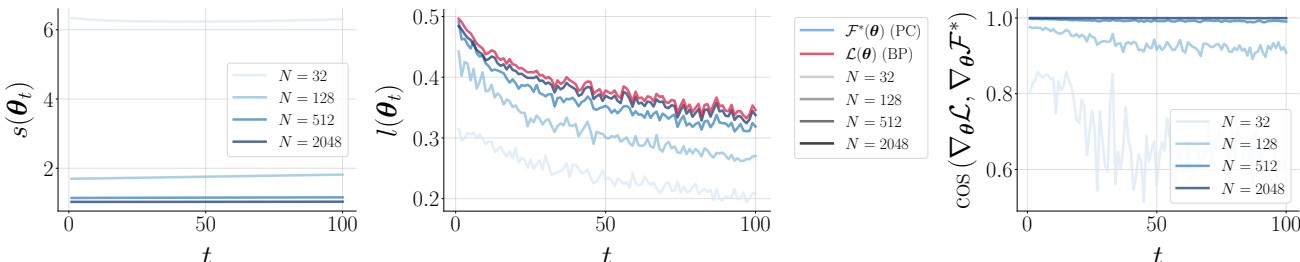

*Figure A.11.* **Depth slice ($L = 32$) results of Figure A.6.** See also the next figure.

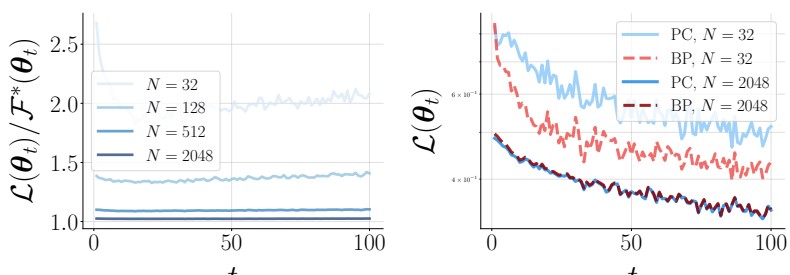

*Figure A.12.* **Additional results for Figure A.11.**

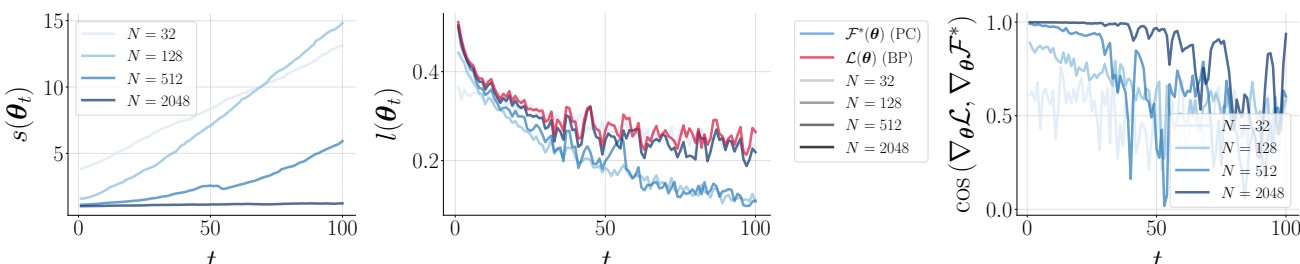

*Figure A.13.* **Results of Figure A.11 for MLPs.** Depth slice ($L = 32$) results of Figure A.8. As reflected in Figure A.8, PC starts to diverge from BP at sufficiently large depth, even at much larger width, because of the MLP forward pass instability with depth (§3.2). See also the next figure.

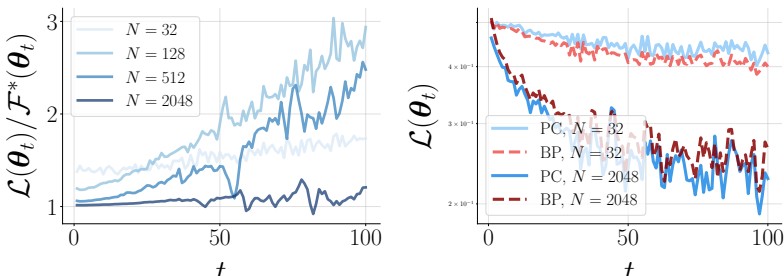

Figure A.14. **Additional results for Figure A.13.**

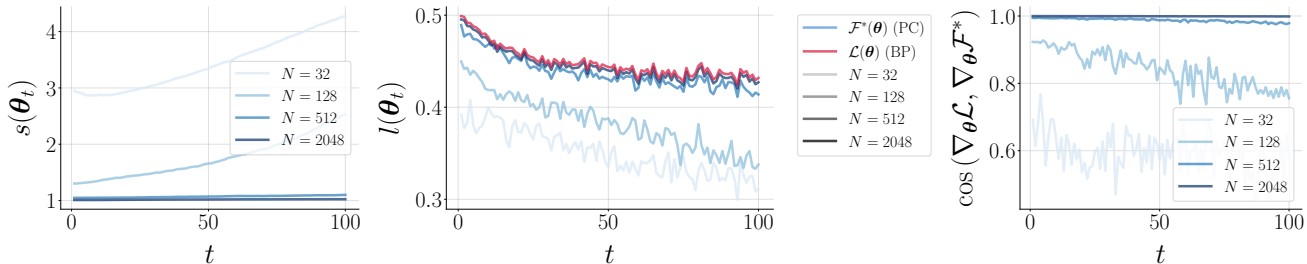

Figure A.15. **Similar results of Figure 2 on CIFAR-10.** We trained 16-layer MLPs of varying widths $N$ on CIFAR-10 using SGD and the mean-field parameterisation (see Table 2). As in Figure 2, we observe: (i) that $s(\theta) \to 1$ as $N \to \infty$ (*Left*); (ii) that the equilibrated energy (Eq. 5) converges to the MSE loss (Eq. 1) (*Middle*); and (iii) that the PC gradients converge to the BP gradients (*Right*). See §A.8 for more details.

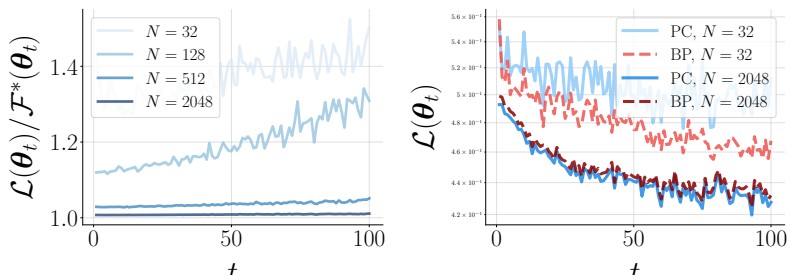

Figure A.16. **Additional results for Figure A.15.**

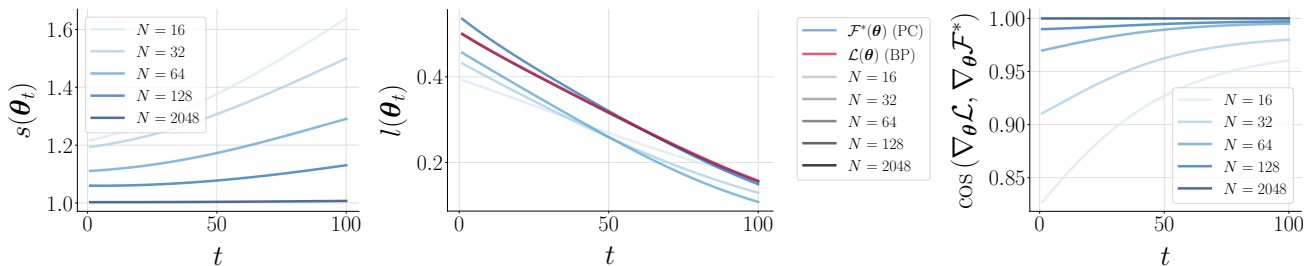

Figure A.17. **Results of Figure 2 for linear residual networks.**

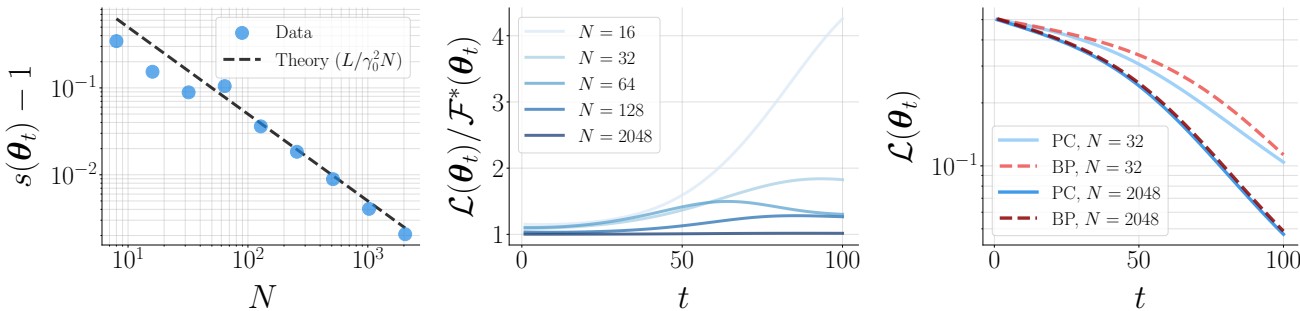

*Figure A.18.* **Additional results for Figure 2.** (*Left*) Comparison between the empirical and theoretical equilibrated energy rescaling $s(\boldsymbol{\theta})$ minus 1, verifying Eq. 16. Note that in this case the depth $L$ and output scaling $\gamma_0$ (see Eq. 11) are constants with respect to the width and so do not affect the asymptotic scaling. (*Middle*) Ratio between the MSE loss (Eq. 1) and the equilibrated energy (Eq. 5) during training, for different network widths $N$. We see that the ratio converges to one as $N \rightarrow \infty$. (*Right*) MSE loss for a network trained with PC and BP for selected widths, showing convergence with large $N$.

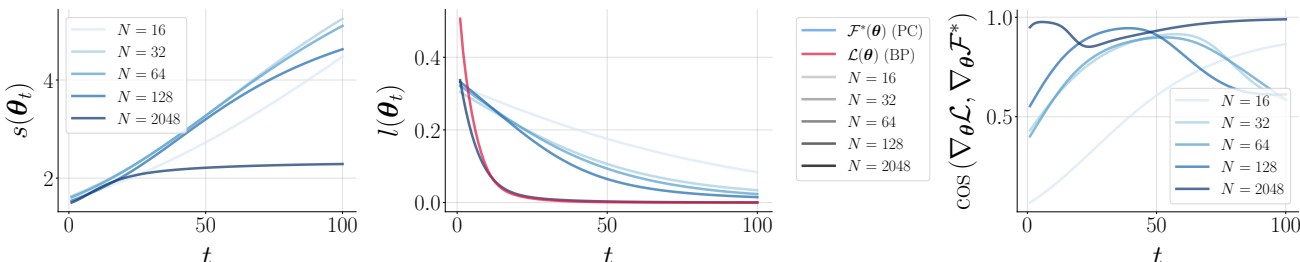

*Figure A.19.* **Results of Figure 2 for the SP.** In contrast to the results of Figure 2 with the mean-field parameterisation (see Table 2), (i) the rescaling $s(\boldsymbol{\theta})$ (Eq. 6) does not converge to one with the width (*Left*), (ii) the equilibrated energy (Eq. 5) does not converge to the MSE loss (Eq. 1) (*Middle*), and (iii) PC generally computes different gradients from BP (*Right*). Note that the gradients align for the widest networks because both the equilibrated energy and the loss are close to zero.

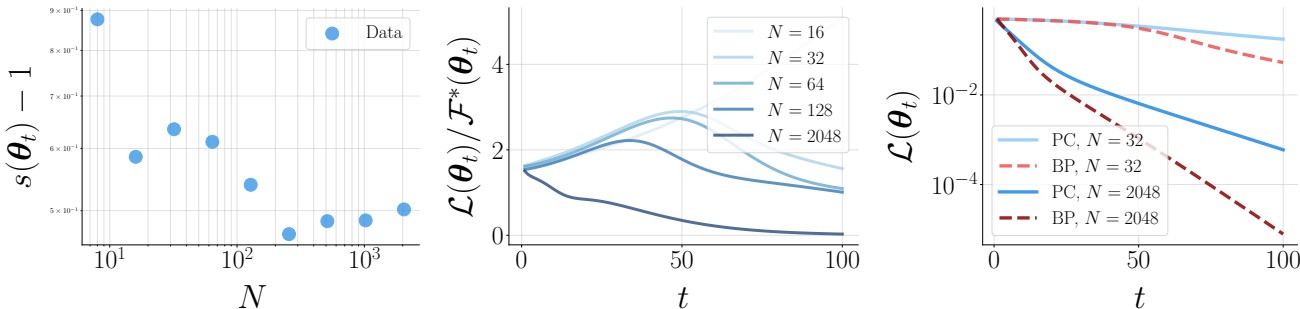

*Figure A.20.* **Results of Figure A.18 for the SP.** In contrast to comparable results with the mean-field parameterisation (Figure A.18), for the SP we observe (i) that the equilibrated energy rescaling $s(\boldsymbol{\theta})$ (Eq. 6) appears to be roughly constant with the width or $\Theta_N(1)$ (*Left*), and (ii) that there is no general correspondence between the equilibrated energy and the BP MSE loss at any width (*Middle & Right*).

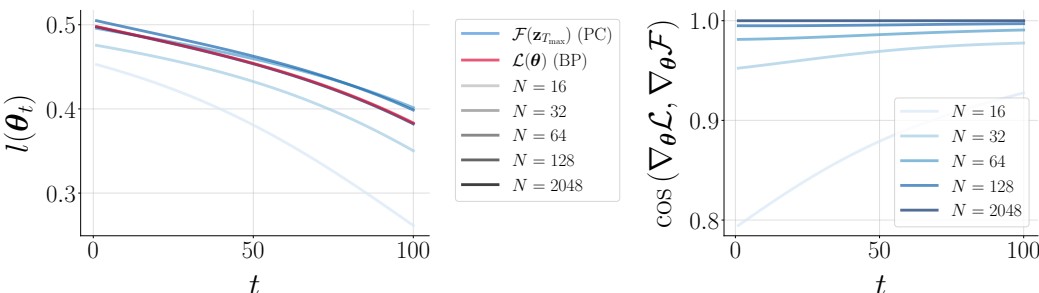

*Figure A.21.* **Results of Figure A.1 for Tanh MLPs.**

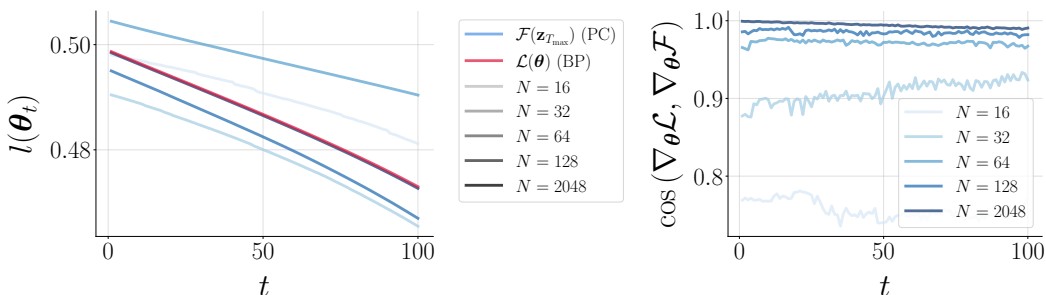

*Figure A.22.* **Results of Figure A.21 for ReLU.**

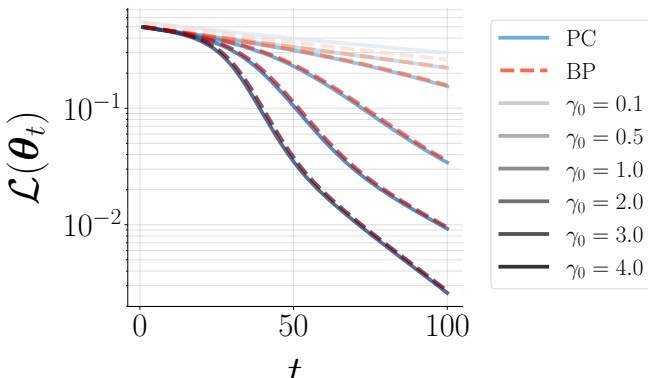

*Figure A.23.* **Results of Figure A.3 for linear residual networks.** As in Figure A.3, we observe that larger values of $\gamma_0$ (Eq. 11) are associated with faster (or richer) learning dynamics.

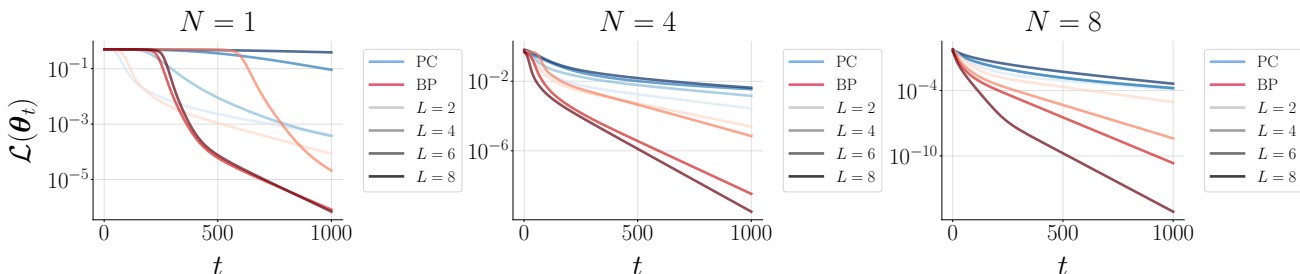

*Figure A.24.* **Results of Figure A.4 for linear residual networks.** As noted in §3.2, the fast PC saddle-to-saddle regime does not exist for residual networks (vs MLPs) since they effectively shift the position of the origin saddle.

