# OpenReview forum: "On the Infinite Width and Depth Limits of Predictive Coding Networks"
_ICML.cc/2026/Conference — ICML 2026 regular_

### Official Review · Reviewer_q5Ys · 2026-03-11

**Soundness:** 4
**Presentation:** 4
**Significance:** 3
**Originality:** 4
**Overall Recommendation:** 5
**Confidence:** 4

**Summary:**

This paper provides a theoretical characterization of the behavior of the predictive coding algorithm for training deep neural networks at large width and large depth. This algorithm alternates between an equilibration phase where the activity of neurons is optimized subject to boundary conditions on inputs and outputs, and an update step where parameters are updated using the equilibrated activity. Similar to the maximum update parameterization $\mu$P desiderata, the authors work out constraints on the initialization and layerwise learning rates to allow stable feature learning at infinite width. The paper focuses on the linear case where the activation function is the identity. In this case, the authors show that the equilibrated energy $\mathcal F^\star(\theta) = \frac{1}{s(\theta)} \mathcal{L}(\theta)$ is simply a rescaled version of the loss function. They find the scale $s(\theta) = 1 + \Theta( L / N)$ which indicates that the PC bias (compared to GD) accumulates with depth $L$ but decreases with width $N$.

**Compliance With Llm Reviewing Policy:**

Affirmed.

**Final Justification:**

The reviewers answered all of my questions and I maintain my support of this paper's acceptance.

**Key Questions For Authors:**

1. I suspect it is theoretically possible to analyze the alternating dynamics of neural activity and plasticity updates for nonlinear networks using the DMFT tools the authors invoke to study the infinite width limit. This could be potentially useful as it would enable the claims in earlier parts of the paper to carry over to the nonlinear case (at least asymptotically in the $N \to \infty$ limit). Have the authors thought about attempting this?

Roughly the idea would be to track at each timestep $t$ of the weight dynamics, iterations for the activity equilibration over $n = \{0,1,2,... \}$ which has the form

$z_{n+1}^\ell(t) = z_{n}^\ell(t) - \beta \left(  z^{\ell}_n(t) -  \frac{1}{\sqrt N} W^{\ell-1}(t) \phi(z^{\ell-1}_n(t) ) \right) + \frac{\beta}{\sqrt N} \text{diag}(\dot\phi(z^\ell_n(t)) W^\ell(t)^\top  \left(  z^{\ell+1}_n(t) -  \frac{1}{\sqrt N} W^\ell(t) \phi(z^\ell_n(t))  \right)$

Expanding out the weight dynamics $W(t) = W(0) + \text{low-rank updates}$ we could then isolate the dependence of the initial random weights and derive DMFT equations for both timescales of the PC dynamics. These wouldn't need to be simulated per se, but could let you state a general result for the existence of width limit for nonlinear PC networks.

The analysis would keep $n$ fixed.  At fixed iteration count $n$ there could be an additional bias in PC compared to exact BP which could account for the need for large $\beta$ in the Figure 4.

2. A paper that could be relevant that is currently missed is this work on how different learning rules (feedback alignment, hebbian learning, etc) can be characterized at infinite width https://arxiv.org/abs/2210.02157. PC offers another nice example of infinite width theory providing crisp predictions about different learning rules.

3. Could the authors clarify a bit their discussion on the depth >> width regime? From the text it sounds like PC is thought to outperform GD in this regime. Yet, the authors then state that PC training should become slower in this regime due to $s \sim 1 + \Theta(L/N)$. First, do the authors think that comparable performance could be achieved by rescaling learning rates $\eta \to \eta \times s$? Second, in my understanding PC is an approximation of GD, so would it not be surprising if PC actually outperformed GD?

**Limitations:**

Yes, the authors do a good job discussing the limitations of the present study.

**Strengths And Weaknesses:**

***Strengths***

**Originality** This paper studies an original problem (scaling limits for PC training in deep networks). It makes a unique connection between the bioplausible learning rules literature and recent works on parameterization design (how to set initialization and learning rates as one varies width and depth). The findings are very novel and shed light on the behavior of PC in larger networks.

**Soundness** The authors' derivations are straightforward and detailed. They also compare their theoretical predictions with experiment. While I did not check all details in their entirety, I checked many derivations in the Appendix which appeared to be correct.

**Significance** Establishing that for width >> depth, the PC algorithm converges to backpropagation is quite a nice theoretical result. This shows that overparameterization through width scaling enables a reduction in approximation error between a local/bioplausible learning rule and exact backpropagation which could not be plausibly implemented in multilayer networks. The work also demonstrates that potential speedups for PC networks that have been observed for $L \gg N$ in non-residual networks are generically unstable in the large network regime, providing some clarification on where speedups compared to BP can be found.

***Weaknesses***

**Analytical Results Rely on Linearity** The primary weakness of this work is that most of the core theoretical results are only proved in the case of linear activation functions. While this makes sense from the context of the equilibration dynamics having a unique fixed point, it reduces the scope of some claims. Section 5 provides empirical evidence of convergent behavior of PC at large width. The agreement between the gradients computed with PC and BP improves with the step size $\beta$ for the activity dynamics.

---

> ### Author Rebuttal · Authors · 2026-03-27
>
> We thank the reviewer for the very positive and thoughtful review. Below we answer their questions in turn.
>
> **1. I suspect it is theoretically possible to analyze the alternating dynamics of neural activity and plasticity updates for nonlinear networks using the DMFT tools the authors invoke to study the infinite width limit. This could be potentially useful as it would enable the claims in earlier parts of the paper to carry over to the nonlinear case (at least asymptotically in the  limit). Have the authors thought about attempting this?**
>
> This is a very interesting suggestion which we did not think about. In full disclosure, we started this study thinking that we would find a ***different*** set of stable and feature-learning parameterisations for PC than BP, and that we would then use DMFT tools to characterise this regime. This was no longer needed once we found that the “optimal” parameterisations for PC and BP are the same (since the DMFT equations would also be the same or very similar). However, it never occurred to us to leverage DMFT to study the infinite width limit of the non-equilibrated energy, also accounting for the activity dynamics. We therefore find the reviewer’s suggestion novel and very likely worth pursuing.
>
> **2. A paper that could be relevant that is currently missed is this work on how different learning rules (feedback alignment, hebbian learning, etc) can be characterized at infinite width https://arxiv.org/abs/2210.02157. PC offers another nice example of infinite width theory providing crisp predictions about different learning rules.**
>
> We thank the reviewer for pointing out this paper, which we missed. We will incorporate it into the paper, both to further motivate our analysis and to compare results with these other learning rules.
>
> **3. Could the authors clarify a bit their discussion on the depth >> width regime? From the text it sounds like PC is thought to outperform GD in this regime. Yet, the authors then state that PC training should become slower in this regime due to $s \sim 1 + \Theta(L/N)$. First, do the authors think that comparable performance could be achieved by rescaling learning rates $\eta \to \eta \times s$? Second, in my understanding PC is an approximation of GD, so would it not be surprising if PC actually outperformed GD?**
>
> First, it is worth clarifying (in answer to the reviewer’s last question) that ***PC is not meant to be an approximation to GD***. Unlike many other bio-plausible learning rules such as equilibrium propagation and feedback alignment, PC is not derived to approximate the BP loss gradient. Instead, PC is derived as a variational inference algorithm over a hierarchical Gaussian model. For this reason, there is no a priori reason to think that the energy weight gradients are an approximation of the BP loss gradients. Indeed, as alluded to in the introduction, work has shown that the weight gradients computed by PC are closer to second-order (Alonso et al., 2022) and even higher-order methods (Innocenti et al., 2024), such that one GD step on the PC energy can be seen as a higher-order GD step on the BP loss.
>
> This brings us to the discussion of the depth >> width regime in Section 3.2, which is mainly based on the results of Innocenti et al. (2024), replicated in the Appendix (see A.7.2). In brief, they showed that, under the standard parameterisation (SP), for PC the escape time from each saddle in the so-called "saddle-to-saddle regime" is fast and independent of depth, compared to BP. This is because the equilibrated energy effectively ***reshapes*** degenerate saddles of the loss into more benign ones. The reason why this saddle-to-saddle regime is connected to depth >> width is because for SP, this aspect ratio effectively initialises the weights closer to the origin. So, in a way, the result that PC converges to BP in the limit studied is not the best one could hope for, given its connection to higher-order methods.
>
> **Weakness: Analytical Results Rely on Linearity**
>
> Regarding this limitation, we would just like to point out that, in response to another reviewer, we tested our main result (i.e. convergence to BP under $\mu$P for width >> depth) on nonlinear convolutional neural networks and transformers. Rather surprisingly, we find that the result holds robustly on both architectures, trained on different datasets including ImageNet. This of course does not change the scope of our theoretical results; we just mention this to further support our claims beyond the linear case.
>
> **References**
>
> Alonso, N., Millidge, B., Krichmar, J., & Neftci, E. O. (2022). A theoretical framework for inference learning. Advances in Neural Information Processing Systems, 35, 37335-37348.
>
> Innocenti, F., Achour, E. M., Singh, R., & Buckley, C. L. (2024). Only strict saddles in the energy landscape of predictive coding networks?. Advances in Neural Information Processing Systems, 37, 53649-53683.

---

> > ### Author Rebuttal · Reviewer_q5Ys · 2026-04-01
> >
> > I thank the reviewers for their detailed responses. I better understand the depth >> width discussion. I maintain my positive score.

---

### Official Review · Reviewer_XiZb · 2026-03-13

**Soundness:** 4
**Presentation:** 4
**Significance:** 3
**Originality:** 3
**Overall Recommendation:** 5
**Confidence:** 4

**Summary:**

This paper analyzes predictive coding networks (PCNs) in the infinite width and depth limits. The main result is that for linear networks, the set of width and depth stable feature learning parameterizations for PC is identical to those for BP (the muP family). Under any such parameterization, the equilibrated PC energy F*(theta) = L(theta)/s(theta) converges to the BP loss as s(theta) -> 1, with s(theta) = 1 + Theta(L/N). Experiments on nonlinear networks support the theory when inference converges. The paper also shows that the one known regime where PC outperforms BP (fast saddle-to-saddle dynamics for L > N) is inherently unstable with both width and depth.

**Compliance With Llm Reviewing Policy:**

Affirmed.

**Final Justification:**

My main concern was regarding novelty/significance, as I thought the paper was mathematically sound, and highly relevant to those in the PC literature, but as originally submitted the nuances of significance over prior work were not clear to me.

The rebuttal comment

> By contrast, we show that, IF one would like to satisfy the $\mu$P desiderata, then necessarily $\mu$P is the only stable and feature-learning parameterisation for PC. So our result is of much stronger practical impact than Ishikawa et al.’s (2024), especially paired with the negative result about the faster learning regime.

is something I agree with upon a closer read. As such, I think the paper stands to be a keystone result in current PC work. Speculatively, the current paper may motivate future researchers to consider novel and creative variations on local-learning / PC that would overcome the limitations that have now been clarified by the authors. This positions the present paper more strongly than I had originally assessed.

**Key Questions For Authors:**

1. For nonlinear networks, do you have any analytical insight into whether s -> 1 persists, or is the correspondence purely empirical?
2. Given the paper's implications that PC converges to BP under stable parameterizations, and the one regime where it differs is unstable, how do you view the practical outlook for PC? A more direct discussion would strengthen the paper.

**Limitations:**

yes

**Strengths And Weaknesses:**

### Strengths
- The theoretical analysis is clean. The core derivation in the main text easy to follow. i.e. checking each muP desideratum against the equilibrated energy gradient flow and showing the PC-specific terms are always subleading at O(N^{-1/2}). The appendix provides full derivations, also easy to follow.
- The paper unifies some previously disconnected results: Ishikawa et al 2024 on width-stable PC parameterization, Innocenti et al. 2025 on depth parameterization (aka muPC), and the PC -> BP convergence conditions from Millidge's line of work. Having these cast in a single framework is valuable.
- That the primary PC advantage is unstable is an important negative result.

### Weaknesses
- Probably the main weakness is around novelty and originality. This largely comes across as follow up work to Ishikawa and muPC. Maybe it's more accurate to say the paper is novel (provides new results) but incremental. The main extensions here are a slightly more general parameterization family following the Bordelon and Pehlevan 2022 setting, and extension to depth via residual networks. The extenstion to depth is useful but arguably technically straightforward.
- All theoretical results concern linear networks. The nonlinear experiments are limited to small-scale settings (N=2048, L<=16, CIFAR-10). Given the stated motivation is scaling PCNs, I maybe expect more thorough exploration but perhaps the authors have a justification for this.
- The practical message is somewhat circular? The paper shows that "good" parameterizations for PC are identical to BP's, and under these parameterizations PC converges to BP. If PC converges to BP under any sensible parameterization, and the one regime where PC differs is unstable, what is the practical motivation for using PC? I think framing the narrative as a strong negative result for PC would be perfectly fine as it may help motivate others to explore different local learning methods or different variations of PC.
- The inference convergence component of the story is underexplored. Section 5 identifies that reaching activity equilibrium becomes much harder for deeper PCNs. But the paper only addresses this empirically. The entire theoretical framework assumes exact convergence, so understanding when and why this breaks down seems important.
- Theorem 2 is stated as covering "any width-stable and feature-learning parameterization" of linear residual networks, but Appendix A 5 fixes the width parameterization to mean-field before doing the depth analysis. The theorem statement is broader than the proof, unless I am missing something.

---

> ### Author Rebuttal · Authors · 2026-03-27
>
> We thank the reviewer for their detailed review and constructive criticism. We address their concerns and questions below.
>
> **Probably the main weakness is around novelty and originality. This largely comes across as follow up work to Ishikawa and muPC. Maybe it's more accurate to say the paper is novel (provides new results) but incremental. [...].**
>
> We find reviewer’s assessment mostly fair. However, we would like to make two points. First, our theoretical results differ significantly from the comparable result of Ishikawa et al. (2024). Specifically, Corollary 4.3 of Ishikawa et al. (2024) shows that there ***exists*** a parameterisation where linear PCNs implement BP or GD. By contrast, we show that, ***IF*** one would like to satisfy the $\mu$P desiderata, then ***necessarily $\mu$P is the only stable and feature-learning parameterisation for PC***. So our result is of much stronger practical impact than Ishikawa et al.’s (2024), especially paired with the negative result about the faster learning regime.
>
> Second, Innocenti et al. (2025) did not provide a theoretical justification of their “$\mu$PC” parameterisation of residual networks beyond the stability of the forward pass and did not use the correct learning rate scaling for Adam, as we discuss in A.2.
>
> **All theoretical results concern linear networks. The nonlinear experiments are limited to small-scale settings (N=2048, L<=16, CIFAR-10). Given the stated motivation is scaling PCNs, I maybe expect more thorough exploration but perhaps the authors have a justification for this.**
>
> This was due to the scaling of the number of steps for convergence of the inference dynamics with the depth. However, we take the reviewer’s point, and so also ran experiments with deeper residual networks, nonlinear CNNs and even transformers. We find that convergence to BP under $\mu$P for width >> depth holds robustly across all architectures, trained on different datasets including ImageNet, optimisers, and loss functions. We will add these results to the paper if accepted.
>
> **The practical message is somewhat circular? [...].**
>
> We agree that the current writing of the paper could be made clearer on the practical implications for PC. Since this is related to the last question of the reviewer, we answer it below.
>
> **The inference convergence component of the story is underexplored. [...].**
>
> We agree with the reviewer that this issue is important but largely beyond the scope of this work. Since we find that our results are highly consistent with those of Innocenti et al. (2025), who performed a theoretical and empirical analysis of the inference landscape, we mainly rely on their results. If the paper is accepted, we will expand on our discussion in Section 5 with these results. We would also be happy to include empirical results on the conditioning of the inference landscape in the appendix, if the reviewer thinks that this would strengthen this section.
>
> **Theorem 2 [...] statement is broader than the proof, unless I am missing something.**
>
> The specific type of width-stable and feature-learning parameterisation considered does not affect the results of the depth analysis (since they are equivalent in terms of the desiderata they satisfy). We now state this explicitly in Appendix A.5 and thank the reviewer for pointing out the imprecision.
>
> **1. For nonlinear networks, do you have any analytical insight into whether s -> 1 persists, or is the correspondence purely empirical?**
>
> For nonlinear networks, there is no rescaling $s(\theta)$, since the energy in general has no unique inference solution that can be plugged back into it to derive an analogous “equilibrated energy”. So in the absence of a theory for the nonlinear case, “all one can do” is to look at the numerical correspondence between the energy and the loss, and their gradients.
>
> **2. Given the paper's implications that PC converges to BP under stable parameterizations, and the one regime where it differs is unstable, how do you view the practical outlook for PC? A more direct discussion would strengthen the paper.**
>
> Overall, we think that our results provide hard constraints on what parameterisations are scalable with PC, but still allowing for potential practical applications, and leaving the door open for other notions of stable parameterisations. First, the result that the only “better” regime for PC is unstable is clearly negative. Second, the result that the only stable parameterisation is essentially one where PC converges to BP could be of practical impact, if ways of performing faster inference are found, on standard or (perhaps more likely) analog hardware. This is because the weight updates of different layers can be parallelised in time with PC. Finally, we do not want to overstate our results to say that there are ***no other notions*** of stable parameterisations (with different desiderata) for PC where it does not converge to BP. We will integrate these points into the paper if accepted.

---

> > ### Author Rebuttal · Reviewer_XiZb · 2026-04-02
> >
> > I thank the authors for the clear response. In particular, I now understand better the novelty of this submission from previous works.
> >
> > > We agree with the reviewer that this issue is important but largely beyond the scope of this work ... We would also be happy to include empirical results on the conditioning of the inference landscape in the appendix, if the reviewer thinks that this would strengthen this section.
> >
> > I agree that this is beyond the scope of the work. I do not think the extra empirical results are required here. Your clarification was helpful.
> >
> > > Overall, we think that our results provide hard constraints on what parameterisations are scalable with PC
> >
> > Thank you, I think this clarifying. It may be that I missed this nuance in the wording of the original submission, but being explicit about these points (at least to me) seem helpful in framing this work in the context of current PC work.
> >
> > I will raise my score to a 5.

---

### Official Review · Reviewer_56jd · 2026-03-13

**Soundness:** 4
**Presentation:** 4
**Significance:** 3
**Originality:** 3
**Overall Recommendation:** 5
**Confidence:** 3

**Summary:**

This paper studies various parameterizations of predictive coding networks (PCNs) in the the infinite width and depth limits. The authors show that, generally, in linear and residual networks the width- and depth-stable feature-learning parameterizations for PCN is the same as for BP. Additionally, the authors show that the PC energy with equilibrated activities converges to the BP loss and the same gradients as BP in a regime where the model width is much larger than the depth. Various experiments show that these results hold in practice for deep nonlinear networks in the case activations in PCNs reach an equilibrium.

**Compliance With Llm Reviewing Policy:**

Affirmed.

**Key Questions For Authors:**

1. What would you say are the practical implications of your theoretical results? Is it that the mu-pc initialization is superior to the standard initializations used for PCN?

2. What are the general conclusions we can make about scaling PCNs given your results? Is it that mu-pc is the only method that will allow for stable scaling? Or do your results suggest there are there still difficulties with scaling PCNs?

**Limitations:**

Yes there are limitation and impact sections.

**Strengths And Weaknesses:**

Strengths:
- Paper is very clearly written and is easy to follow.
- Diagrams used to explain algorithms and math concepts are helpful
- Plots are clear and easy to understand
- Paper focuses on little studied area of PCN networks, i.e., developing a theoretical understanding of initialization regimes, and provides novel results.
- These results seem somewhat useful in understand how to best initialize PCN networks and in understanding what architectures yield updates the most similar to BP.
- Significant further detailed empirical results and mathematical explanation in appendix.

Weakness:
- Paper is limited to theoretical results and empirical studies  aimed to support theoretical results. Practical significance seems somewhat limited.

---

> ### Author Rebuttal · Authors · 2026-03-26
>
> We thank the reviewer for their overall positive review. We answer their questions below.
>
> **1. What would you say are the practical implications of your theoretical results? Is it that the mu-pc initialization is superior to the standard initializations used for PCN?**
>
> As we note in **Implications** (Section 6.1), our theoretical results have 3 main implications:
> * Because the set of width- and depth-stable parameterisations for PCNs turn out to be the same as for BP (Theorems 1-2), and the standard parameterisation (SP, e.g. relying on PyTorch initialisation) is width-unstable for BP, then ***SP is also not scalable for PC***. This is a practically important result since the majority of previous works, including those trying to scale PCNs (e.g. see Pinchetti et al, 2024), use exactly the SP.
> * Related, as highlighted in Takeaway 2, the only regime where PC has been clearly shown to provide advantages over BP (in terms of faster learning convergence) relies on the SP (Innocenti et al., 2024). ***This potentially better regime is therefore also not scalable***.
> * ***IF*** one would like to satisfy the same desiderata as $\mu$P—which are very non-restrictive and have enabled stable and efficient scaling of BP—then our results ***necessarily*** imply that the mean-field/$\mu$P parameterisations are the ***only*** scalable parameterisations for PC, in the sense of being numerically stable and non-trivial at large width and depth.
>
> We of course strictly show all of the above only for linear PCNs, but the experiments on nonlinear networks—which we now supplement with results on convolutional neural networks and transformers, based on feedback from another reviewer—clearly show that the results hold for the nonlinear case.
>
> The reviewer also asks if our results mean that the “$\mu$PC” initialisation is superior to the standard initialisations used for PCNs. The answer depends on what is meant by “$\mu$PC”. First of all, we would like to clarify that a parameterisation is defined not only by scalings of the initialisation, but also by scalings of the learning rate and pre-activations, as summarised in Table 1 and explained in detail in Section 2. Second, the term $\mu$PC was introduced by Innocenti et al. (2025) to refer to a slightly different depth parameterisation for BP, excluding the prescribed learning rate scaling for Adam (as we discuss in A.2). In our work, we show that this scaling should in fact be included (since it is necessary for BP). However, if by "$\mu$PC" it is strictly meant what we define as any width- and depth-stable feature-learning parameterisation, namely the mean-field and $\mu$P parameterisations, then the answer is "yes": $\mu$PC is superior to the standard parameterisation.
>
> **2. What are the general conclusions we can make about scaling PCNs given your results? Is it that mu-pc is the only method that will allow for stable scaling? Or do your results suggest there are there still difficulties with scaling PCNs?**
>
> As explained in the summary of implications above, ***IF*** one would like to satisfy the same desiderata as $\mu$P—which seems reasonable given its practical success for BP—then “yes”, $\mu$PC (i.e. the mean-field and $\mu$P parameterisations for PC) is indeed the ***only*** method for stable scaling with both width and depth. As we note at the end of Section 6.2, it could be interesting to investigate whether there exists other notions of stable and feature-learning parameterisations (requiring different desiderata), although this seems challenging given that the $\mu$P desiderata seem to be highly non-restrictive.
>
> If the paper is accepted, we will use the extra space to clarify the practical implications of our work.
>
> **References**
>
> Innocenti, F., Achour, E. M., Singh, R., & Buckley, C. L. (2024). Only strict saddles in the energy landscape of predictive coding networks?. Advances in Neural Information Processing Systems, 37, 53649-53683.
>
> Innocenti, F., Achour, E. M., & Buckley, C. $\mu$PC: Scaling Predictive Coding to 100+ Layer Networks. In The Thirty-ninth Annual Conference on Neural Information Processing Systems.
>
> Pinchetti, L., Qi, C., Lokshyn, O., Emde, C., M'Charrak, A., Tang, M., ... & Salvatori, T. Benchmarking Predictive Coding Networks--Made Simple. In The Thirteenth International Conference on Learning Representations.

---

> > ### Author Rebuttal · Reviewer_56jd · 2026-03-31
> >
> > The authors adequately clarified the practical implications over their research and committed to clarifying these implications in the paper.

---

### Official Review · Reviewer_DQzr · 2026-03-13

**Soundness:** 3
**Presentation:** 4
**Significance:** 3
**Originality:** 3
**Overall Recommendation:** 5
**Confidence:** 2

**Summary:**

Predictive coding is an alternative to standard backprop which is behind almost every modern deep learning method. Predicitve coding network (PCN) can be difficult to train however. This paper studies BP-inspired reparameterisations, meaning applying specific width-dependent and depth-dependent scaling exponents to the network's architecture to ensure stable training dynamics across scales.

The primary theoretical finding is that for linear networks with equilibrated activities, the exact set of width- and depth-stable, feature-learning parameterizations for PCNs is mathematically identical to those derived for backpropagation. Under these stable parameterizations, as long as the network width is substantially larger than its depth, the equilibrated PC energy rigorously converges to the BP mean squared error loss, meaning PC ultimately computes the exact same weight gradients as BP. Furthermore, the theoretical analysis proves that the "fast saddle-to-saddle" convergence regime—previously identified as a unique computational speed advantage of PC—is inherently unstable when scaling both the width and depth of the network.

Empirically, the authors demonstrate that this theoretical convergence of PC gradients to BP gradients accurately holds in practice for deep nonlinear models, such as those using Tanh and ReLU activation functions, provided the width is much larger than the depth. This practical convergence, however, strictly requires the PC inference phase to successfully reach a numerical activity equilibrium. Reaching this required equilibrium becomes progressively more difficult in deeper networks due to the ill-conditioning of the inference optimization landscape, which restricts practical scalability and requires careful tuning of the activity optimization step sizes.

**Compliance With Llm Reviewing Policy:**

Affirmed.

**Final Justification:**

The rebuttal addressed all my concerns and reinforced my previous positive assessment.

**Key Questions For Authors:**

1. Since the practical convergence to BP gradients relies heavily on the inference phase reaching numerical activity equilibrium, how does the computational overhead required to reach this equilibrium scale with depth in practice? At what specific scale does the wall-clock time required for inference optimization render the PCN approach practically uncompetitive with standard BP implementations?

**Limitations:**

Yes. The biggest limitation as the authors noted is that the analysis depends on studying the equilibrated energy. I appreciate the careful discussion in Section 6.2 on this.

**Strengths And Weaknesses:**

Soundness:

It makes sense to me that to study how PCNs can be scaled up that we turn to existing results on BP-parametrisations in infinite-width and infinite-depth regimes. The theory looks rigorous and the experiments have a good amount of stress testing including testing on nonlinear.

Presentation:

I did not know anything about predictive coding networks. I found the Background Section 2 very well written and clear.

Significance:

The primary significance of this work lies in establishing a rigorous theoretical framework that unifies and explains previous empirical heuristics. By formalizing the infinite-width and depth limits, the authors provide a mathematical justification for the specific parameterizations necessary to stabilize deep predictive coding networks. Furthermore, the paper's "negative" results offer substantial scientific value; by formally demonstrating that the fast saddle-to-saddle convergence regime is asymptotically unstable , and that the inference landscape becomes increasingly ill-conditioned at scale , the authors establish clear, theoretically grounded boundaries on the current practical scalability of predictive coding.

Originality:

The originality of this work stems from formally applying infinite-width and infinite-depth limit analyses—methodologies traditionally reserved for standard backpropagation—directly to the equilibrated energy landscape of Predictive Coding Networks.

---

> ### Author Rebuttal · Authors · 2026-03-26
>
> We thank the reviewer for an excellent summary and positive assessment of our work. We answer their questions below.
>
> **1. Since the practical convergence to BP gradients relies heavily on the inference phase reaching numerical activity equilibrium, how does the computational overhead required to reach this equilibrium scale with depth in practice?**
>
> This question was studied in detail in Innocenti et al. (2025), as discussed in Sections 5 and 6, and we answer based on their findings. Empirically, Innocenti et al. (2025) showed that the inference landscape of linear and nonlinear residual networks grows ill-conditioned with the depth, both at initialisation and during training. Based on their results, the condition number seems to scale superlinearly with the depth at initialisation, and growing at a rate proportional to the depth during training. Theoretically, they suggested (although did not prove) that this pathological scaling is due to the block tridiagonal structure of the Hessian of the energy with respect to the activities, which grows sparser with the depth. If the paper is accepted, we will expand on our discussion in Section 5 with these results.
>
> We agree with the reviewer that, given that convergence to BP holds only at an activity equilibrium (under the studied parameterisations and for width >> depth), the practical convergence of the inference dynamics is an important issue that would benefit from further study.
>
> As a side note, it is thought that such iterative inference dynamics—which are a feature of many other energy-based algorithms such as equilibrium propagation—could be performed significantly faster by directly instantiating the system in (e.g. neuromorphic) hardware, rather than simulating it as in digital. As an encouraging example, Aifer et al. (2024) showed that stochastic processors can perform matrix inversion significantly faster than standard digital methods, with the speed-up growing with both the matrix dimension and condition number. For these reasons, we believe that hardware and software co-design is an important future direction for the field.
>
> **2. At what specific scale does the wall-clock time required for inference optimization render the PCN approach practically uncompetitive with standard BP implementations?**
>
> Based on previous estimates for digital hardware such as GPUs (Salvatori et al., 2022), the point at which a single training step with PC (involving both inference and learning) becomes slower than BP is roughly when the number of inference steps is larger than the number of layers. This comes from the fact that weight gradients can be computed in parallel across layers with PC, due to the layer-wise structure of the energy. This is in contrast to backprop, where the gradient computation of any layer needs to wait for that of the previous layers. However, this advantage of PC is lost if the sequential inference dynamics take more steps than number of layers. We again emphasise that these estimates assume digital hardware. If the paper is accepted, we will use the extra space to add this point of discussion to Section 6.
>
> **References**
>
> Aifer, M., Donatella, K., Gordon, M. H., Duffield, S., Ahle, T., Simpson, D., ... & Coles, P. J. (2024). Thermodynamic linear algebra. npj Unconventional Computing, 1(1), 13.
>
> Innocenti, F., Achour, E. M., & Buckley, C. $\mu$PC: Scaling Predictive Coding to 100+ Layer Networks. In The Thirty-ninth Annual Conference on Neural Information Processing Systems.
>
> Salvatori, T., Song, Y., Yordanov, Y., Millidge, B., Sha, L., Emde, C., ... & Lukasiewicz, T. A Stable, Fast, and Fully Automatic Learning Algorithm for Predictive Coding Networks. In The Twelfth International Conference on Learning Representations.

---

> > ### Author Rebuttal · Reviewer_DQzr · 2026-04-03
> >
> > Thank you for your clarifications. I'm glad to see all reviewers are unanimously positive.

---

### Decision · Program_Chairs · 2026-04-30

**Decision:**

Accept (regular)

**Comment:**

Authors study feature learning parameterizations for predictive coding networks in the infinite width and depth limits. These parameterizations turn out to be identical to backpropagation parameterizations.

Predictive coding is an important biologically plausible alternative to backpropagation, but as other alternatives suffers from experimentation at large scale. Therefore, attempts at scaling these networks is valuable and may shed light onto credit assignment in the brain. All reviewers agree that this is an important work.

Reviewers brought several technical points as well as questions about scope and novelty. The rebuttal by authors addressed all these questions. All reviewers vote for accept. I recommend acceptance.